# Plant roots increase both decomposition and stable organic matter formation in boreal forest soil

Bartosz Adamczyk [1,2,3,4], Outi-Maaria Sietiö [2,3], Petra Straková[4,5], Judith Prommer[6], Birgit Wild[6,7,8,9], Marleena Hagner[10], Mari Pihlatie [1,2,11], Hannu Fritze[4], Andreas Richter [6] & Jussi Heinonsalo[1,2,3,12]

Boreal forests are ecosystems with low nitrogen (N) availability that store globally significant amounts of carbon (C), mainly in plant biomass and soil organic matter (SOM). Although crucial for future climate change predictions, the mechanisms controlling boreal C and N pools are not well understood. Here, using a three-year field experiment, we compare SOM decomposition and stabilization in the presence of roots, with exclusion of roots but presence of fungal hyphae and with exclusion of both roots and fungal hyphae. Roots accelerate SOM decomposition compared to the root exclusion treatments, but also promote a different soil N economy with higher concentrations of organic soil N compared to inorganic soil N accompanied with the build-up of stable SOM-N. In contrast, root exclusion leads to an inorganic soil N economy (i.e., high level of inorganic N) with reduced stable SOM-N build-up. Based on our findings, we provide a framework on how plant roots affect SOM decomposition and stabilization.

[1] Department of Agricultural Sciences, University of Helsinki, PO Box 66Helsinki, Finland. [2] Institute for Atmospheric and Earth System Research (INAR), University of Helsinki, Helsinki, Finland. [3] Department of Microbiology, University of Helsinki, PO Box 66Helsinki, Finland. [4] Natural Resources Institute Finland, PL 2, 00791 Helsinki, Finland. [5] Department of Forest Sciences, University of Helsinki, PO Box 27Helsinki, Finland. [6] Department of Microbiology and Ecosystem Science, University of Vienna, Althanstr. 14, 1090 Wien, Austria. [7] Department of Earth Sciences, University of Gothenburg, Gothenburg, Sweden. [8] Department of Environmental Science and Analytical Chemistry, Stockholm University, Stockholm, Sweden. [9] Bolin Centre for Climate Research, Stockholm University, Stockholm, Sweden. [10] Natural Resources Institute Finland, Tietotie 2, 31600 Jokioinen, Finland. [11] Viikki Plant Science Centre (ViPS), University of Helsinki, Helsinki, Finland. [12] Finnish Meteorological Institute, Climate System Research, Helsinki, Finland. Correspondence and requests for materials should be addressed to B.A. (email: bartosz.adamczyk@luke.fi)

Boreal forests are key components of the global carbon (C) cycle due to their high C storage and enormous potential for C sequestration into soil organic matter (SOM)[1]. Nevertheless, with climate change SOM decomposition might increase shifting boreal forests from C sinks to C sources, thereby accelerating global warming[1]. The mechanisms behind accumulation and stabilization of SOM in boreal forest soils are poorly understood but essential for predicting C stocks in a future climate[2]. The boreal forest C storage is tightly linked to the nitrogen (N) cycle[3], which is characterized by the binding or complexation of a large fraction of soil N to other soil compounds such as minerals and polyphenols, resulting in low N availability[4–8]. Thus forest soil N is to a large extent present in chemically stable form[9], with the remainder in labile form (easily available N forms, dissolved inorganic and organic N) or retained in living organisms, e.g. mycorrhizal fungi[10].

Transformations of boreal forest SOM are driven by a complex network of interactions among soil microorganisms, including ectomycorrhizal and ericoid mycorrhizal (EEM) fungi, saprotrophic fungi, bacteria[11] and plant roots[12]. Plant roots support microorganisms in the rhizosphere, i.e. in the narrow soil zone surrounding roots[11], with easily available C, which may stimulate microbial activity and thereby increase SOM decomposition (referred to as the rhizosphere priming effect)[13–15]. Interactions between different microbial guilds can further influence SOM decomposition rates; for instance, EEM fungi may suppress fungal saprotrophs, thereby decreasing SOM decomposition and carbon dioxide ($CO_2$) production (a phenomenon known as the Gadgil effect)[16,17]. In addition to soil microorganisms, soil fauna may enhance the SOM decomposition by fragmentation and processing dead organic matter into a more available form for microbes[18]. An emerging view underlines the role of microorganisms not only in SOM decomposition but also in SOM stabilization[19–21], i.e. the transformation of SOM into more stable forms. According to the microbial carbon pump concept, microorganisms metabolically process plant residues and generate biomass, and microbial residues are stabilized in soil (in vivo turnover)[19], via multiple mechanisms including reactions with minerals and soil aggregates[22]. Alternatively, microbial enzymes may stabilize plant residues by transforming them into less-available forms (ex vivo modification)[19]. Overall, EEM plants may thus accelerate SOM decomposition through the rhizosphere priming effect or decelerate decomposition (e.g. via the Gadgil effect) and increase the formation of stable SOM[12,23,24] (e.g. via the microbial C pump)[19]. The balance between microbial priming and soil C increase has been proposed to regulate the stable soil C pool[19,25]. Although crucial for modelling responses of SOM stocks to land use and climate change[26], an in-depth mechanistic understanding of SOM transformations is still lacking.

This study aims to elucidate the mechanisms underlying SOM transformations and improve our understanding of its controls, especially those related to N pools. The classic hypothesis is formed that roots and associated microorganisms would stimulate microbial biomass growth and activity (priming) and consequently SOM decomposition rates including stable SOM to obtain N for plants and vice versa the lack of roots would result in lower decomposition rates (hypothesis 1). Alternatively, this root-induced priming effect is at some point exceeded by the build-up of stable SOM-N from decomposition-released organic C and N inputs (hypothesis 2). This would lead to stable soil C formation especially in treatment with roots providing significant amounts of C and N inputs.

Here we test these hypotheses in a field experiment where mesh bags filled with unsterilized, homogenized soil organic layer with the natural microbial community were returned into the organic layer of the boreal pine forest soil and collected after one, two and three growing seasons. Three mesh sizes were compared that excluded the ingrowth of plant roots as well as mycorrhizal hyphae (1 µm mesh size treatment), permitted the ingrowth of hyphae but not of roots (50 µm mesh size treatment) or permitted the ingrowth of both hyphae and roots (1000 µm mesh size treatment). We compare SOM loss and stable SOM-N formation, soil chemistry and fauna, microbial community structure and activities (enzymatic activity and microbial N transformations) between the mesh size treatments, focussing specifically on N pools and fluxes that may be tightly linked to C storage in boreal forests[27]. Based on our field experiment, we provide a framework on the role of plant roots in SOM decomposition and stabilization. Plant roots not only accelerate SOM decomposition but also increase the chemically stable N pool. On the other hand, exclusion of plant roots leads to increased levels of inorganic N (IN) and reduced build-up of stable soil N, as well as decreased decomposition rates due to a lack of rhizosphere priming support from plants. Thus our field study sheds light on the mechanisms behind plant root-driven SOM transformations, increasing the systems understanding needed for improving forest soil management to enhance SOM accumulation.

## Results and discussion

**Decomposition and microbial decomposers.** In our 3-year experiment, chemical and biological soil properties (Fig. 1) were significantly affected by both time and mesh size ($P < 0.001$, $R^2 = 0.490$ overall effect for different years and $P < 0.05$, $R^2 = 0.328$ overall effect for different mesh sizes, with permutational multivariate analysis of variance (PERMANOVA)). The 1000 µm

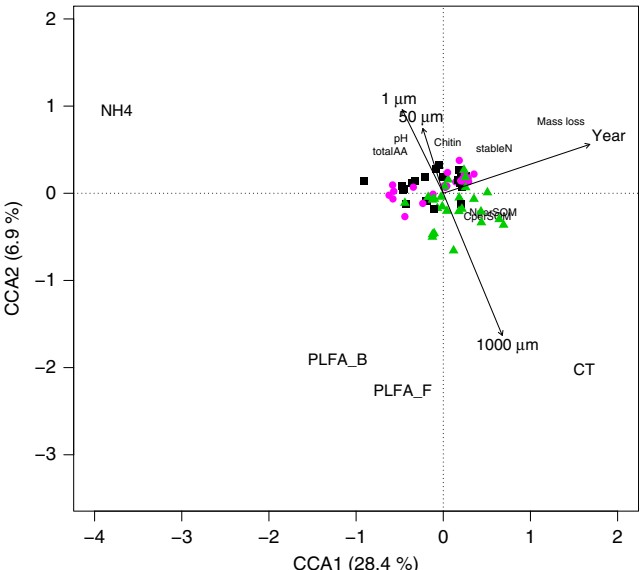

**Fig. 1** Effect of the treatments and time on chemical and biological soil properties. Partial constrained correspondence analysis (pCCA) illustrates the effect of the three treatments and experimental time (year) on the chemical and biological compositions of the mesh bags. The CCA1 axis explains 28.4% ($P \le 0.001$, $F = 31.36$) of the data variability, and the CCA2 axis explains 6.8% ($P \le 0.001$, $F = 7.59$) of the data. The different mesh treatments are represented with a green triangle (1000 µm mesh), magenta circle (50 µm mesh) and black square (1 µm mesh). The variables responsible for the separation of the different samples in the analysis have been marked on the diagram with the following abbreviations: total AAs total free amino acids, CT condensed tannins, stableN stable SOM-nitrogen (N) pool, SOM soil organic matter, NH₄ ammonium, PLFA_B bacterial phospholipid fatty acids (PLFA), PLFA_F fungal PLFA

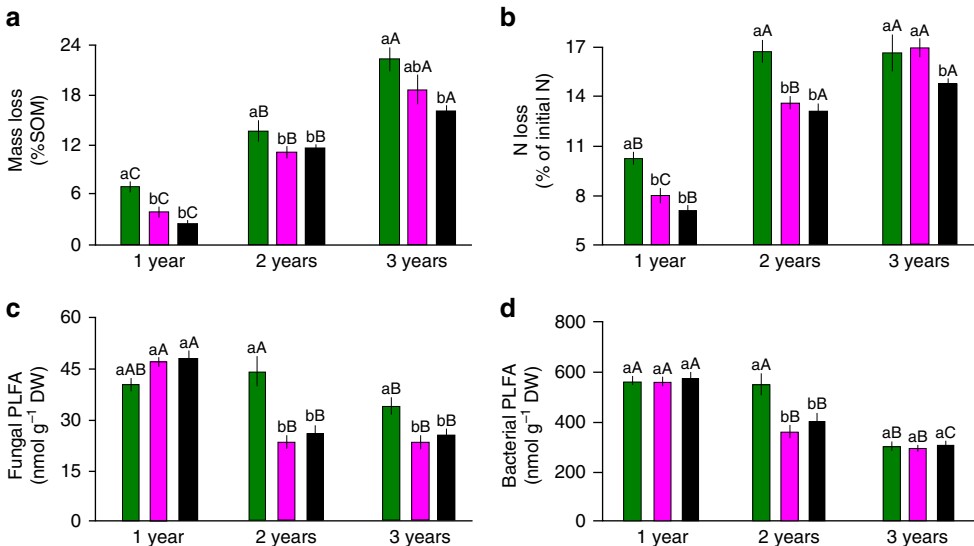

**Fig. 2** Soil decomposition and microbiology: **a** Soil organic matter (SOM) loss, **b** N loss, **c** PLFA fungal biomarkers, **d** PLFA bacterial biomarkers. The different mesh sizes are represented with different colours (from left to right): 1000 μm (green), 50 μm (magenta), and 1 μm (black). The given values are the means of 24 replicates. Significant differences ($P < 0.05$) between treatments within 1 year are indicated by different letters, and the differences between different years for the same treatment are indicated by capital letters. The error bars represent ±s.e.m. Source data are provided as a Source Data file

treatment that permitted root ingrowth showed the highest SOM and total N loss throughout all years ($P < 0.01$; Fig. 2a, b). In the last year, the fungal hyphal treatment (50 μm) also tended to accelerate SOM and N losses ($P < 0.09$ and $P < 0.05$, respectively, Fig. 2a, b). These results confirm our first hypothesis that microorganisms supported by plant C increase SOM decomposition, compared to microorganisms lacking plant C input (1 μm treatment). The positive effect of plant roots on SOM decomposition rate falls in line with previous studies on rhizosphere priming effect[13,14,28], as well as with a meta-analysis where decomposition was significantly higher in the rhizosphere than in the bulk soil[15]. Phospholipid fatty acid (PLFA) analysis showed that microbial community composition was significantly affected by time since exposure ($P < 0.001$ and $R^2 = 0.516$ with PERMANOVA) and by treatment ($P < 0.05$ and $R^2 = 0.077$ when the sampling year was set as grouping factor with PERMANOVA; Supplementary Fig. 1). Fungal PLFA markers became more dominant with time in the 1000 μm treatment comparing to the other treatments in the same year (Fig. 2c), whereas bacteria showed higher abundance in 1000 μm treatment only during the second year ($P < 0.05$) (Fig. 2d). The highest SOM decomposition in the 1000 μm treatment may therefore be ascribed to increased activity of fungal decomposers supported by plant-derived carbon input (i.e. rhizosphere priming[15]). In line with that, combined fungal DNA and ergosterol results after 3 years showed increased colonization in 1000 and 50 μm bags by ectomycorrhizal and saprotrophic fungi compared to 1 μm bags (Supplementary Fig. 2). Similarly, also root ingrowth into 1000 μm increased with time of incubation (Supplementary Table 1). We did not find major differences in soil mesofauna and microfauna between treatments (Supplementary Table 2) and this suggests that the effect of plant roots and associated microorganisms explains most of the changes in SOM decomposition.

In accordance with the higher SOM decomposition rate in the 1000 μm treatment, we expected to see enhanced microbial activity and a decrease in the stable SOM-N (the chemically stable N pool), which would be in agreement with the N mining hypothesis of rhizosphere priming[13,14] and hypothesis 1. Although fungal abundance was higher in the 1000 μm than in

the other treatments, we did not find higher soil $CO_2$ production or soil enzymatic activity. Soil respiration was only marginally higher in the 1000 μm treatment compared to other treatments, and enzymatic activity was largely at the same level, with significantly higher activities only for the 50 μm treatment after 2 years ($P < 0.01$; Supplementary Figs. 3 and 4). We additionally conducted a follow-up experiment to test for treatment effects on microbial N transformation activities. After harvesting the mesh bags from the soil, roots were removed and the soil homogenized, and gross rates of protein depolymerization, N mineralization as well as microbial amino acid and ammonium uptake were quantified using [15]N-pool dilution assays[29]. We did not detect consistent differences between the treatments (Supplementary Fig. 5), contradicting hypothesis 1, as microbial activities should be enhanced owing to priming. Since some of the above-mentioned results (i.e. soil respiration, enzyme activities) are derived from single point measurements after the autumn harvest, and are most sensitive to moisture changes, these results should be, however, treated with caution. In contrast, the general soil chemical and microbial indicators (e.g. SOM loss, PLFA) better reflect changes over the entire incubation period, giving more reliable long-term estimates of belowground processes.

**Formation of stable SOM**. Although SOM loss was the highest in the 1000 μm treatment, also here the stable SOM-N pool showed the strongest increase among the three treatments, by 23% after 3 years ($P < 0.01$, Fig. 3a). This result contradicts hypothesis 1 and suggests that the presence of plant roots did not only enhance net SOM decomposition but also simultaneously the formation of stable SOM-N. Characterization of soil from mesh bags with Fourier-transform infrared spectroscopy (FTIR) revealed differences between treatments mainly in polyphenols and polypeptides (Supplementary Figs. 6 and 7). The concentrations of condensed tannins (CTs) and FTIR absorbance values representing polyphenols were greatest in the 1000 μm treatment (Fig. 3b, c), and CTs were significantly correlated with the stable SOM-N ($r = 0.680$, $P < 0.01$, Pearson correlation, Fig. 3d). In addition, FTIR absorbance values representing polypeptides as

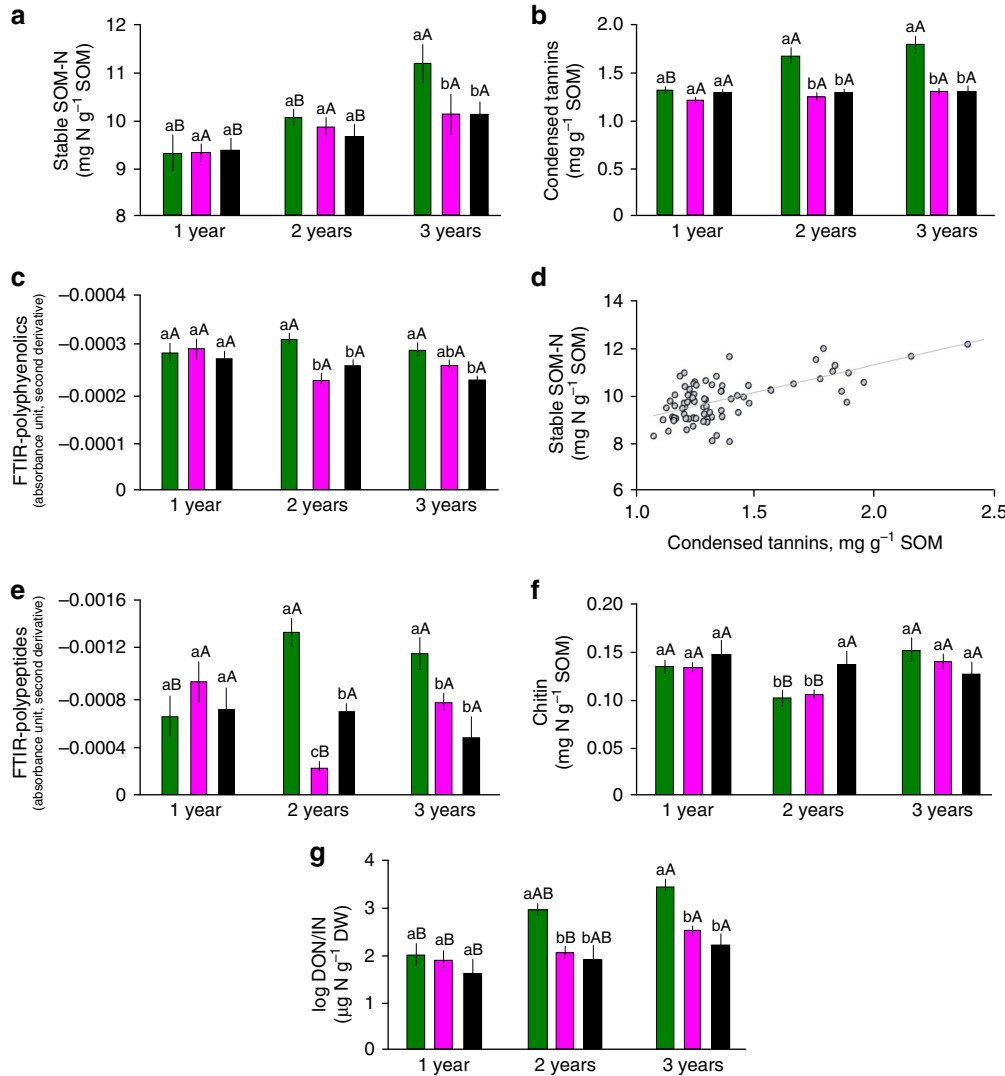

**Fig. 3** Soil chemistry and correlations with condensed tannins: **a** stable SOM-N pool, **b** condensed tannins, **c** Fourier-transform infrared (FTIR) polyphenolics (sums of 1512, 1421 and 1388 infrared (IR) bands), **d** correlation between stable SOM-N vs CT plotted for the whole set of data, **e** FTIR polypeptides (sums of 1668 and 1543 IR bands) (see Supplementary Figs. 5 and 6), **e** chitin concentration, **f** chitin concentration, **g** dissolved organic N/ inorganic N (DON/IN) ratio (logarithmic values). The different mesh sizes are represented with different colours (from left to right): 1000 μm (green), 50 μm (magenta) and 1 μm (black). The given values are the means of the 24 replicates (except 9 for FTIR and chitin). More results characterizing the soil are given in Supplementary Figs. 3–6. Significant differences ($P < 0.05$) between treatments within 1 year are indicated by different letters, and the differences between different years for the same treatment are indicated by capital letters. The error bars represent ±s.e.m. Source data are provided as a Source Data file

well as concentrations of chitin were also greatest in the 1000 μm treatment (Fig. 3 e, f) and correlated with CTs (for FTIR polypeptides $r = 0.698$, $P < 0.036$, chitin for third year only $r = 0.750$, $P < 0.001$, Pearson correlation). We here propose that this significant increase in the stable SOM-N resulted from the formation of complexes with root-derived CTs, i.e. polyphenols able to form complexes with proteins[30–32], and other organic N compounds, e.g. chitin[4,5]. These results are in line with our earlier paper, in which we showed that CTs stabilize fungal necromass in boreal forest soil[33,34]. Thus we suggest the formation of organic N–CT complexes as one of the mechanisms behind the stabilization of SOM-N in the presence of plant roots, which is in agreement with build-up of SOM from root- and fungal-derived biomass[20,27,35,36], and supports our hypothesis 2. In our experiment, formation of stabile SOM-N reached 5 mg N g$^{-1}$ SOM after 3 years. Assuming that complexation of proteins and chitin

with CTs is responsible for stabile SOM-N formation and knowing that C:N ratio of complexes is about 6:1[37], we can extrapolate it into 30 mg of C stabilized after 3 years. Taking into account that fine root turnover may be higher than previously assumed[38] and that fine roots provide significant amount of CTs to the soil[35], this mechanism might play a significant role in SOM stabilization in boreal forest soils. In addition, increased fungal biomass could directly decrease N availability through N immobilization[23] and increase stabilization of SOM via recalcitrant melanin production[39,40]. Therefore, those potential mechanisms of SOM build-up could explain how ectomycorrhizal trees decrease N availability in boreal forest ecosystems and drive soil C accumulation.

Going further, it is important to consider the role of stable SOM-N formation through reactions with CTs in the N cycle. The complexation of proteins by tannins has been proposed to

induce a shift from mineral- to organic-dominated N cycling, i.e. that an increasing fraction of dissolved N released from litter is in organic rather in than mineral forms[7]. In line with this concept, we observed the highest dissolved organic N (DON)/inorganic N (IN) ratio in the 1000 μm treatment ($P < 0.05$; Fig. 3g), which reflects natural forest conditions. Lower concentrations of IN in this treatment may emerge not only from lower N mineralization but also from higher IN uptake by plants roots. Previous work at our study site in Hyytiälä also demonstrated a tight N cycle that was dominated by organic N and proposed that most of the N assimilated into tree biomass originated from SOM decomposition, with negligible amounts of IN in the soil[41].

**Soil nitrogen economy.** The combined results suggest that plant roots with associated microorganisms (1000 μm) promoted an organic soil N economy, i.e. the domination of organic N forms and stable SOM-N build-up (Fig. 4). In contrast, the artificial treatments where ingrowth of roots (50 μm) and roots as well as fungal hyphae (1 μm) were excluded showed an inorganic soil N economy with the accumulation of IN (Fig. 4), potentially promoting N losses by leaching and denitrification. This N economy observed for boreal forests cannot be simply extrapolated to temperate and tropical forests where soil N stock per area ($kg\,N\,m^{-2}$) is about three times lower than in boreal forests[24]. Moreover, formation of stable SOM-N may be significantly affected by quantity and quality of soil components. For instance, the clay content of boreal forest soils is almost three times lower than that of temperate forests and up to ten times lower than that of tropical forests[24]. Similarly, also

mycorrhizal component can play a crucial role, as it was shown that soil in ecosystems with ecto- and ericoid mycorrhiza (like boreal forests) contains 70% more C per unit N than soil in ecosystems dominated by arbuscular mycorrhizal plants[24], suggesting different mechanisms of soil C storage.

We have to be aware that the mesh bag-based experimental design may suffer from some caveats. For example, in addition to the loss of SOM from mesh bags due to decomposition and uptake, C loss may partially emerge from leaching of dissolved organic carbon (DOC) into deeper soil horizons[42]. The translocation of organic C to mineral soils may also stabilize organic matter, which is responsible for the formation of decadal and centurial old C pools in boreal forest soils[43]. The DOC pool may also be susceptible to leaching from the soil to surface and ground waters, although as the DOC outflow from undisturbed forest soils is generally low[44], we assumed that also in our experiment DOC leaching is of minor effect. Similarly to the trenching method[45], mesh bag manipulations can also potentially be affected by differences in moisture and limitation of soil fauna entrance. However, in our study the moisture content and soil fauna abundance did not show major differences between treatments (Supplementary Tables 1 and 2). All in all, these caveats did not have substantial effect on our results.

**Implications for biogeochemical predictions.** We provide a framework on the role of plant roots in SOM decomposition and stabilization based on a field experiment in boreal forest soils (Fig. 4). Plant roots promote organic N economy not only with

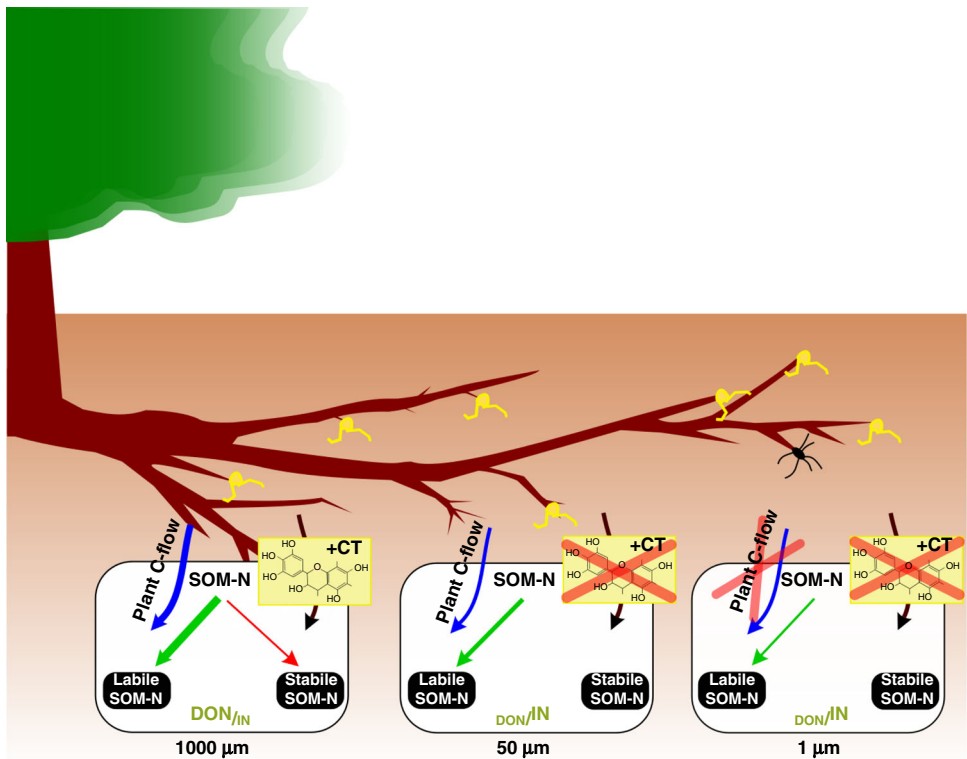

**Fig. 4** Framework of SOM-N transformations: organic N economy under natural conditions (1000 μm), inorganic N economy for exclusion of roots (50 μm), as well as roots and fungal hyphae (1 μm). Organic N economy (i.e. the domination of organic N) includes also build-up of stable SOM due to stabilization of organic N compounds with plant-derived tannins. Exclusion treatments (50 and 1 μm) led to an inorganic N economy (i.e. increased concentration of inorganic N forms) with reduced SOM build-up. The arrow thickness is proportional to the magnitude of the net flux. CT condensed tannins (shown as a monomer), SOM soil organic matter, DON dissolved organic N, IN inorganic N. The sizes of DON and IN are proportional to their concentrations

accelerated SOM decomposition but also with stabilization of SOM-N. On the other hand, exclusion of plant roots leads to increased levels of IN (IN economy) and reduced build-up of stable soil N, as well as decreased SOM decomposition rates due to a lack of rhizosphere priming support from plants. The framework provided by our study may facilitate an improved understanding of boreal forest soil responses to global change and improve predictions of biogeochemical effects of possible shifts in SOM accumulation and decomposition. The incorporation of this framework into other forest ecosystems appears to be the next step to understand mechanisms controlling the soil C sequestration. However, extrapolation to other ecosystems than boreal forests may require significant modifications according to differences in plant CT contents and soil processes that are affected by, for example, mycorrhizal type (ectomycorrhiza vs arbuscular mycorrhiza) and its effect on soil carbon content[24] or differences in the C cost of N acquisition[46]. Finally, additional studies are required to test how major global environmental changes (e.g. N deposition, climate warming) alter interactions between EEM fungi and free-living microbial decomposers[24] and, consequently, change soil C storage.

## Methods

**Experimental design.** Soil for this study was collected in the vicinity of the SMEAR II station of Helsinki University at Hyytiälä (61°84'N, 24°26'E) in southern Finland (see ref. [47] for site details). The soil was a haplic Podzol, and Scots pine (*Pinus sylvestris* L.) was the dominating tree species. The soil was taken from the organic layer (Ofh), visible roots were removed and the soil was homogenized and sieved through a 4-mm mesh. Such prepared soil was placed in mesh bags (corresponding to 14.2 g dry weight per bag) with different mesh size to form 3 types of treatment: mesh size 1000 μm did not limit fine root and hyphal in-growth, 50 μm mesh excluded roots but not fungal hyphae, and 1 μm excluded root and also fungal penetration[48]. Mesh bags in 24 replicates per treatment (1, 50, 1000 μm) were placed on topmost mineral horizons in the soil organic layer relative to horizon position at Hyytiälä station in May 2013. Mesh bags were collected in 3 September campaigns in 2013, 2014 and 2015. Samples were transported to the laboratory at +4 °C and after removal of roots divided for different measurements (see below). Enzyme and DNA analyses, $^{15}$N pool-dilution and KCl extractions were conducted using fresh soil, the rest of the samples was frozen at −20 °C, and later freeze-dried and ground to a fine powder using a ball-grinder (2000–230 Geno/Grinder, Spex Sample Pred, US).

**Total C, N, SOM and pH.** Total soil C and N contents were determined with an elemental CN analyser (LECO, Michigan, USA). SOM content was measured as loss on ignition at +550 °C and soil pH in a soil–water suspension of 1:2.5.

**Phospholipid fatty acids.** Phospholipid extraction and PLFA analysis were carried out as described earlier[49]. Briefly, between 1 and 1.5 g of fresh soil was freeze-dried and extracted with a chloroform:methanol:citrate buffer mixture (1:2:0.8) and the lipids were separated into neutral lipids, glycolipids and phospholipids on a silicic acid column. The phospholipids were subjected to mild alkaline methanolysis, and the fatty acid methyl esters were detected by gas chromatography (flame ionization detector) using a 50-m HP-5 (phenylmethyl silicone) capillary column. Helium was used as a carrier gas. Oven temperature was 50 °C and raised at a rate of 30 °C min$^{-1}$ to 160 °C, then at a rate of 2 °C min$^{-1}$ to 270 °C and finally kept constant for 5 min at 270 °C. Peak areas were quantified by adding methyl nonadecanoate fatty acid (19:0) as an internal standard before methanolysis. The sum of the following PLFAs was considered to be predominantly of bacterial origin and chosen as an index of bacterial biomass (PLFA, bacterial biomarkers): i15:0, a15:0, 15:0, i16:0, 16:1ω9, 16:1ω7t, i17:0, a17:0, 17:0, cy17:0, 18:1ω7, and cy19:0. The quantity of 18:2ω6 was used as an indicator of fungal biomass (PLFA, fungal biomarker)[49,50].

**Fungal DNA analysis.** Total DNA was extracted from 50 mg of soil using the NucleoSpin® Soil DNA Extraction Kit (Macherey-Nagel GmbH&Co). All the extracted DNA samples were further purified with the PowerClean® Pro DNA Clean Up Kit (MO BIO Laboratories, USA) following the manufacturer's instructions. From the extracted total DNA, fungal ITS2 regions were sequenced using Illumina® MiSeq at the Institute of Biotechnology (University of Helsinki). The general read quality of raw ITS2 reads was checked with the FastQC software (http://www.bioinformatics.babraham.ac.uk/projects/fastqc/), and adapter and barcode sequences were trimmed away with the Cutadapt software[51]. The sequence data were further filtered and clustered to operational taxonomic units using mothur[52] and following the workflow described in ref. [53]. For

identification, fungal sequences were aligned against UNITE database and the obtained fungal taxa were assigned to functional guilds according to the FUN-Guild database[54].

**Fungal biomass.** Fungal biomass was measured through its biomarker, ergosterol, with the classic high-performance liquid chromatography (HPLC) method[50]. Briefly, ergosterol was extracted from 0.25 g soil samples by adding 1 ml cyclohexane and 4 ml 10% KOH in methanol. After 15 min ultrasonic treatment, the samples were incubated at 70 °C for 60 min. After the heat treatment, 1 ml distilled water and 2 ml cyclohexane were added. The tubes were shaken and the top phase was moved to another test tube. The procedure with water and cyclohexane was repeated and the combined extracts were evaporated under N$_2$ gas at 40 °C. The samples were re-dissolved in methanol by heating at 40 °C for 15 min and filtered (0.2-μm PTFE filter). The amount of ergosterol was measured with HPLC (HP Agilent 1100, Hewlett Packard, USA) using a C18 100A reverse-phase column. Ergosterol was analysed using a 10-μl sample injection and isocratic separation with methanol (1 ml min$^{-1}$) and detected at 282 nm.

**Soil fauna.** Soil mesofauna and microfauna were measured using modified sugar flotation method, where soil (2 g) was first sieved through 1-mm mesh, diluted to 100 ml sugar:water suspension (1:1) and then let to settle 2 h before extraction (25-μm sieve) of soil organisms from the suspension[55].

**Ammonium-N, nitrate-N and total free amino acid content.** Fresh soils (5 g) from mesh bags were extracted with 20 ml of 1 M KCl for 80 min on a reciprocal shaker (120 rpm). Extracts were centrifuged, filtered (0.45-μm filters, PALL Corporation) and frozen at −20 °C until analysis. IN (NO$_3$-N and NH$_4$-N) was measured according to ref. [56]. The assay for ammonium-N was based on a modified indophenol method and for NO$_3$-N on the reduction of nitrate by vanadium (III). Soil total free amino acids were measured from KCl extracts using a fluorometric method with *o*-phthaldialdehyde and β-mercaptoethanol, modified as in ref. [57]. Absorbance and fluorescence were measured with a microplate reader (Infinite M200, Tecan, Switzerland).

**Chemically stable and degradable N pools.** We separated soil N pools into a hydrolysable N pool (hereafter called degradable N pool) and non- hydrolysable N pool (hereafter called stable N pool) using acid hydrolysis with methyl sulfonic acid (MSA)[58,59]. Briefly, dried and ground soil was hydrolysed with 4 M MSA in an autoclave (136 °C, 134 kPa; 90 min) in the presence of tryptamine to avoid amino acid degradation. This hydrolysis released the degradable N pool but not the recalcitrant N pool. Extracts were neutralized with KOH and stored at −20 °C before analysis for total dissolved N content (TOC-Vcph/cpn TNM-1, Shimadzu, Japan), which consequently reflects the degradable N pool. The chemically stable pool was calculated as the difference between total soil N content and the degradable N pool.

**Concentration of CTs.** CTs were extracted from 0.2 g ground soil with 70% aqueous acetone, dried, dissolved in water and exposed to a modified acid-butanol assay (proanthocyanidin assay)[60] after extraction from soil with acetone:water (7:3). The method involves the HCl-catalysed depolymerization of CT in butanol to yield a pink-red anthocyanidin product, which was measured spectrophotometrically. As a standard, we used CT extracted from Norway spruce needles, which was profoundly characterized before (see ref. [61] for details).

**Concentration of chitin.** Chitin was measured using HPLC as described by ref. [62]. Briefly, 5 mg of dried and grounded soil was treated with 0.2 N NaOH to remove proteins and amino acids. The soil was hydrolysed with 5 ml 6 N HCl for 7 h at 105 °C. After centrifugation (5000 × g, 10 min), 500 μl of hydrolysate was evaporated to dryness under nitrogen to remove HCl, redissolved in 1 ml of water, evaporated and redissolved again in water and centrifuged (5000 × g, 10 min). The supernatant was then used to measure the glucosamine content (product of chitin hydrolysis in HCl), after derivatization with 9-flourenylmethylchloroformate. Samples were mixed with homocysteic acid (5 μM) as internal standard. Separation and detection were carried out on a Waters HPLC with a 250 × 5 mm$^2$ (I.D.) ODS-hypersil (5 μm) column. The column effluent was directed to a fluorescence detector (excitation 330 nm and emission 445 nm). Results are presented as mg glucosamine per g$^{-1}$ SOM.

**$^{15}$N pool dilution assays.** Gross rates of protein depolymerization, microbial amino acid uptake, N mineralization and microbial ammonium uptake were measured using $^{15}$N pool dilution assays, as described in principle by Kirkham and Bartholomew[63]. For gross rates of protein depolymerization and microbial amino acid uptake, we followed the assay of Wanek et al[29]., with slight modifications. Duplicates of fresh soil (ca. 1 g) were amended with a mixture of $^{15}$N-enriched amino acids (18.75 ng μl$^{-1}$, 500 μl, ≥96 at% $^{15}$N) and incubated for 10 and 30 min, respectively. Samples were then extracted with 20 ml 10 mM CaSO$_4$ that contained 3.7% formaldehyde to stop microbial activity, filtered and loaded on pre-cleaned

cation exchange cartridges. Cartridges were eluted with 3 M $NH_3$, and samples were amended with internal standards (nor-valine, nor-leucine, para-chloro-phenylalanine, 1 µg each per sample), dried, derivatized with ethyl-chloroformate[29] and analysed with gas chromatography–mass spectrometry (Thermo TriPlus Autosampler, Trace GC Ultra and ISQ mass spectrometer, Agilent DB-5 column). Concentrations of alanine, glycine, isoleucine, leucine, phenylalanine, proline, valine, aspartate+asparagine and glutamate+glutamine were quantified against external standard curves, and the corresponding $^{15}N$ contents were calculated from peak areas of light- and heavy-mass fragments[29]. Aspartate and asparagine, as well as glutamate and glutamine, were not distinguished since formaldehyde causes the deamination of asparagine to aspartate and of glutamine to glutamate[29]. We corrected for background concentrations of amino acids and incomplete recovery from the cation exchange cartridges using blanks and a second set of standards that were processed together with the samples throughout the procedure.

For gross rates of N mineralization and microbial ammonium uptake, duplicates of fresh soil (ca. 1.5 g) were amended with $^{15}N$-enriched $(NH_4)_2SO_4$ (0.125 mM, 500 µl, 10 at% $^{15}N$), and incubated for 4 and 24 h, respectively. Samples were then extracted with 15 ml 2 M KCl and filtered through ash-free filter paper. Ammonium was isolated into acid traps[64], and the amount and isotopic composition were measured with elemental analysis–isotope ratio mass spectrometry (CE Instrument EA 1110 Elemental Analyzer, Finnigan MAT ConFlo II Interface, Finnigan MAT DeltaPlus IRMS). We corrected for background concentrations of ammonium using blanks that were processed together with the samples throughout the procedure.

Gross rates of protein depolymerization, microbial amino acid uptake, N mineralization and microbial ammonium uptake were calculated using equations modified after Kirkham and Bartholomew[63]. Microbial N use efficiency was calculated as the proportion of N taken up in the form of amino acids that was allocated to growth and enzyme synthesis and not to N mineralization[65].

**Enzyme measurements**. Extracellular enzymes were recovered from the fresh soil using the filter centrifugation method[66]. Activities of acid phosphatase (EC 3.1.3.2), chitinase (EC 3.2.1.14), β-glucosidase (EC 3.2.1.21), β-glucuronidase (EC 3.2.1.31), β-xylosidase (EC 3.2.1.37), cellobiohydrolase (EC 3.2.1.91) and leucine aminopeptidase (EC 3.4.11.1) were measured using fluorometric substrates (Sigma)[67]. Fluorescence was measured with a Wallac Victor[2] (Perkin Elmer, Inc., USA) plate reader using excitation at 355 nm and emission at 460 nm.

**Fourier transform infrared spectroscopy**. Soil infrared spectra were recorded with a FTIR spectrometer (Shimadzu IRPrestige-21; Shimadzu Corporation, Japan) equipped with a high-sensitivity DLATGS detector with Germanium-coated KBr beam splitter and signal/noise ratio 40,000:1. We used the IRsolution 1.40 software (Shimadzu Corporation). The spectral data within the band range of $4000–400\ cm^{-1}$ were recorded, and 24 scans were averaged for each spectrum with a spectral resolution of $4\ cm^{-1}$. Three milligrams of dry and finely grounded soil from mesh bags were mixed with 300 mg of KBr and pressed into discs (13 mm in diameter) by applying 112 bars pressure for 2 min (hydraulic press, Perkin Elmer). A blank KBr beam splitter was used to adjust to the baseline level before measurements. All analyses were performed in nine replicates.

**Statistical analysis**. To determine the differences between treatment means, a two-way analysis of variance (ANOVA) test was used with time (year) and mesh size as fixed factors. Where necessary, data were transformed (log10 or arcsin for percentage data) to fulfil the assumptions of the ANOVA. The interactions between time (year) and treatment (mesh size) were all significant. The effect of factors (time or mesh size) were determined separately with post hoc Tukey tests. In the figures, significant differences ($P < 0.05$) between treatments within 1 year are indicated by different letters, and the differences between different years for the same treatment are indicated by capital letters. PERMANOVA test was conducted with the adonis function from the vegan package in the R program (R core team)[68] using 999 permutations. In the PERMANOVA, the duration of the experiment was used as explanatory variable. When testing the effect of mesh size with PERMANOVA, duration of the experiment was used as grouping factor. A partially constrained correspondence analysis (pCCA) was performed in the R program with the function cca from the vegan package[68]. Prior to pCCA, cross-correlations between variables were determined with the corvif function, and only variables not significantly correlated with each other were selected for the analysis. All variables were normalized between 0 and 1 with the 'range' option in the decostand function in the vegan package. In the CCA, the effect of the experimental site was partialized by setting it as 'Condition' in the CCA. The statistical significances of the CCA axes and explanatory variables were determined with the anova.cca function. The pCCA ordination was visualized with the plot.cca function and scaled for site scores (scaling = 1).

For the analysis of PLFA patterns, a linear discriminant analysis (LDA) was performed with the MASS package in the R software. Since the sampling year had the most significant effect on the PLFA patterns and values, the heterogeneity between years was normalized between 0 and 1 with the decostand function from vegan[68] prior to performing the LDA, in order to visualize the differences caused by the mesh treatments.

Infrared spectra were transformed by the second derivative (i.e. the Savitzky–Golay method) using Unscrambler (CAMO, Norway) to remove the baseline shift and enhance the spectral information[69]. A principal component (PC) analysis was performed with Canoco 5 (Microcomputer Power, USA). Wavenumbers that were most affected by treatment (best fit in PC ordination space) were selected and assigned, as in a previous study[70]. A sum of absorbance values at 1668 and $1543\ cm^{-1}$ was used as a representative of polypeptides, and a sum of absorbance values at 1641, 1512, 1421 and $1388\ cm^{-1}$ was used as a representative of polyphenolics.

**Reporting summary**. Further information on research design is available in the Nature Research Reporting Summary linked to this article.

## Data availability
The data that support the findings of this study are available on request from the corresponding author (B.A.). The source data underlying Figs. 1–3 and Supplementary Figs. 1–7 are provided as a Source Data file.

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

## Acknowledgements

This research was supported by grants (263858 and 292669) from the Academy of Finland, two grants from the Finnish Cultural Foundation (granted to B.A. and O-M.S.), the Niemi Foundation and the University of Helsinki Doctoral Program of Microbiology and Biotechnology (MBDP; to O.-M.S.). Miikka Olin is acknowledged for help with the chitin measurements.

## Author contributions

B.A., J.H. and A.R. planned and designed the research; B.A., J.H., A.R., O.-M.S., P.S., M.P., M.H., J.P., B.W. and H.F. performed the experiments and data analysis. B.A. wrote the manuscript with help from other co-authors.

## Additional information

**Competing interests:** The authors declare no competing interests.

