## [Peer Review File · Nature Communications]

Reviewers' comments:

Reviewer #1 (Remarks to the Author):

The manuscript "Plant roots increase decomposition but also formation of stable soil organic matter in boreal forest soils" aims to contribute to mechanistically understand SOM transformation and N stabilization in forest humus by studying soil biological and chemical parameters in the absence or presence of roots and ectomycorrhizal fungal hyphae. This is an important and generally interesting scientific aim addressing several concepts related to the role of microbiota in soil organic matter transformation: rhizosphere priming vs. Gadgil effect as well as the emerging of the microbial carbon pump explaining SOM decomposition and stabilization. The broader context is a better understanding of C-transformation in boreal coniferous forest soils – a major C-sink in the northern hemisphere and thus an important factor in global C-budgeting.

The results base on a three year study using a soil mesh-bag experiment with different mesh sizes to prevent ingrowth of hyphae or roots or both. The mesh-bag samples of every year were analyzed using a suite of techniques that allow quantitation of different processes connected to SOM mineralization, turnover and stabilization as well as the link to N-mobilization and stabilization. The major results are that there were clear effects of mesh size on the dynamics and outcome of C-transformations and that the mesh size allowing in-growth of roots resulted in a higher stabilized SOM and organic N than when roots were excluded. The excellent and complementary chemical analyses on C and N-dynamics showed this convincingly. However, there are three major points in the study that I find missing to accept the conclusion that was drawn from this study.

1. My major concern is that the possible role of soil fauna is not considered at all: neither studied nor discussed - although one could easily argue that this study is about the impact of excluding or including soil mesofauna as well as on root exclusion vs. inclusion. Do the authors find the faunal component irrelevant for their concept, and if so why? Literature on the importance of soil (meso)fauna on SOM turnover and N dynamics is easy to access (for example: Setälä, H., Marshall, V. G., and Trofymow, J. A 1996. Influence of body size of soil fauna on litter decomposition and 15N uptake by poplar in a pot trial. *Soil Biol. Biochem.* 28, 1661–1675). Further hints can be found in a recent review Briones MJI 2018. *The Serendipitous Value of Soil Fauna in Ecosystem Functioning: The Unexplained Explained. Frontiers in Environmental Science*, 6(149). Ideally, these missing data / information are included in a revised version.
2. There is no evidence given that ectomycorrhizal hyphae were present in the 10 µm mesh bags. Although I find PLFA data useful for quantitative numbers and in this aspect superior to DNA-based methods, in this specific case, DNA-based (amplicon sequencing) analyses are necessary to proof presence / absence of ectomycorrhizal taxa in the different mesh-size bags. This information could also be used to estimate presence and absence of different functional guilds of fungi i.e. saprotrophs and mycorrhizal fungi.
3. In the same line of thinking I was missing data on root biomass in the 1 mm mesh-bags which would add evidence on the importance of roots.

Apart from the general concerns above, the strong positive aspects of the study are the chemical analytical methods and the results thereof which are of high quality and strongly contributing to new insights in SOM decomposition / stabilization in coniferous forest soils. The presentation of the results in the text and figures is generally very clear and convincing. Please find some detailed remarks on missing information and suggestions for improvements or clarification in the subsequent paragraph:

Abstract:

The abstract is clear and gives the major idea and results. However, the term "mechanistic framework" appears overselling in the light of not providing data on soil fauna or an estimate on ectomycorrhizal vs. saprotrophic fungal components.

Introduction:

line 111: "plant roots" seems too general. Consider writing "plant roots and/or mycorrhizae".

Results and Discussion:

Line 124 onwards and Methods Summary line 270-272: the adonis function from vegan is described in the documentation as "robust alternative to parametric MANOVA". If the adonis function was used here, it would be better to write either "adonis" or "PERMANOVA" as vegan describes it. Instead of F-values, adjusted R² values would be more informative (again if adonis was used here).

Line 127, 169: also for results that were not statistically significant but "tended to..." P-values should be given as an indication how strongly these tendencies are to be rated.

Line 131: Why is this surprising? Is it realistic to assume that many mycorrhizal hyphae would grow into a root-free space? Here DNA analyses on fungi could give an indication on differences in the presence of ectomycorrhizal vs. decomposer fungi (see general comment 2 above).

Line 134: Citation 30: From the short sentence it is not clear which way of C-input via roots is addressed (rhizosphere deposits or dead root mass) and how the findings of Solly et al. on fine root turnover fit in the context of accelerating SOM decomposition in the present experiment.

Line 147: Has root biomass been determined in the 1000 µm mesh-bags? Was root ingrowth comparable among mesh-bags? Please add this information. In addition, biomass and identity of faunal components should be given in this paragraph.

Line 157: the claims regarding microbial community composition are based on PLFA which is fine for quantification of biomass but the resolution at the taxonomic is very low thus preventing any deeper insights into microbial community composition. Maybe this could be worded a bit more modestly: "community structure at the investigated level of PLFA... ()

Line 167: please add which type of correlation was used (Pearson?).

Line 237: "soilC" space is missing: "soil C"

Conclusion:

Apart from my concerns regarding the scientific part of the conclusions, the broader implications (line 210-213) are very generally worded and can be deleted without losing any information. I suggest to rethink the essence of the conclusion.

Methods Summary

For adonis see comment above for line 124 onwards

Supplementary

Supplementary Figure 2: Reference 27 (line 214) should probably be 57 as in lines 292 and 502?

Supplementary Figure 4C: There seems to be a typo in the post-hoc designation a, ab, ab - one "ab" should be "b".

Reviewer #2 (Remarks to the Author):

Adamczyk et al. : Plant roots increase decomposition but also the formation of stable soil organic matter in boreal forest soil.

This study potentially makes a very interesting and valuable contribution to the literature on mechanisms controlling SOM decomposition and stabilization in the soils of boreal forests.

Warming climate is affecting C sequestration process and causing shifts in SOM accumulation and decomposition. Thus we are in urgent need for more empirical studies that provide fundamental step forward in our understanding of mechanisms of SOM stabilization and accumulation processes. I like very much the functional approach of this study, although I suspect there might be some treatment issues (mesh size) that may distort the results.

The design of the experiment and the respective results raised the question: even though the root exclusion treatment (50µm) should allow only the hyphae to grow into the mesh bag, whether there were more of hyphal biomass compared to root and hyphae exclusion treatment (1 µm)? Fig 2c seems to confirm that there was no difference between the treatments (1000 µm, 50µm and 1 µm) during the first year – fungal PLFA was high in all treatments, also roots and fungal hyphae excluding (1 µm) mesh bags (still saprotrophs or old mycorrhizal signal, because the soil was not

sterilized?); during the second and third year the in-growth of fungi into mesh bags with net size 50 μ m and 1 μ m were similar (50 μ m and 1 μ m seem to highly correlate, in Fig 1), while only the latter was meant to prevent hyphal in-growth. My question is can the authors confirm higher hyphal in-growth in root exclusion treatment (50 μ m) or the growth of (mainly) mycorrhizal hyphae decreased without roots during second year similarly to 1 μ m mesh bags?

Line 49: does "exclusion of roots" mean presence of hyphae? This question relates to my questions about design (see above).

Line 51: concentration of organic...what?

Lines 131-134: the statements here relate to my comments above. The authors state that C flow via mycorrhizal hyphae was not restricted in 50 μ m, but fungal PLFA values show lower in-growth since second year. I would ask the authors discuss this issue.

140-141: fungi were quite dominant already in the first year.

Line 184: Fig 2i should be Fig 2j

Lines 194-201: This is important and topic-related section of text, but feels a bit stand-alone here. The experiment was carried out in upper organic layer of the soil and DOC leaching effects are like out of frame, although I agree that DOC leaching from boreal forest soil is marginal and DOC has an important role in stabilization process of SOM in deeper layers. Can the authors make stronger link between their results and DOC leaching?

Line 212: Phrasing is not good: 'increasing the systems understanding' is not clear, what systems?

However, this is very well written manuscript with clear message. Most novel and innovative are the results explaining the mechanisms underlying the stabilization of SOM (line 171-172) via differences in N-economy. And I find the framework presented on Fig 3 to be intriguing for further discussions on soil C cycle. Well done!

Reviewer #3 (Remarks to the Author):

In this manuscript, Adamczyk et al. aim to "elucidate the mechanisms underlying SOM transformations and its controls, focusing on the rhizosphere," specifically in a boreal forest ecosystem context. The authors used a three year field experiment, which involved different sized mesh bags that either excluded roots and hyphae (1 μ m mesh), excluded roots only (50 μ m), or did not exclude roots or hyphae (1000 μ m). They found that decomposition of SOM was elevated in the presence of roots + hyphae (1000 μ m) relative to the two other treatments, while the 'chemically stable N pool' increased in this treatment by 3, relative to the other treatments. They aim to use their results to present a mechanistic framework for how roots affect SOM decomposition and stabilization.

This study is a useful effort to try to untangle the role of roots and hyphae in N cycling (though for C cycling, which the authors also frequently mention, it would have been useful to have more detailed data on soil carbon). The role of roots and hyphae in the cycling of N and SOM is an understudied subject, especially in boreal systems, and it is particularly useful to have data over multiple years using a manipulative field design. The authors measured a wide suite of response variables, to provide a complete story, and these results definitely warrant publication. However, I have some concerns as to how the results are framed in terms of advancing understanding around the emerging idea of microbes for their role in stabilizing and decomposing SOM, as I'm not sure the study exactly addresses this point. Rather, it looks specifically at hydrolysable versus non hydrolysable N, but not the SOM pool. I find the data most compelling to provide a story of N dynamics, and not as much around SOM or C.

Overall, I find the conceptual framework to be a bit of an overreach, based on the results of the experiment. While the results point to some interesting questions to follow up on, I do not feel they provide conclusive mechanistic evidence to advance such a mechanistic framework. For example, it seems to me that a large part of the justification for the framework is based on an observed correlation between condensed tannins and the size of the stable N pool. I would prefer

the paper include more figures/tables that present actual data (at this point, only two data figures are included).

That said, I commend the authors on conducting this important study, and advise that the take-home messages from the study be articulated a little more carefully and clearly – I found the last two paragraphs of the Discussion a bit hard to follow. One broad point is that I would like to see in the Discussion these results discussed in the context of boreal systems, and how different dynamics may play out in other forest ecosystems. Also, I would like to see the authors more persuasively state why these findings in this study are novel, to warrant publication in Nature Communications. The third paragraph of the Introduction states the aim of the paper, but I felt it did not persuasively state why the hypotheses are unique, in the context of prior literature on this subject, and does not give background as to why the questions and approach are very novel. Last, I'd also like to see the authors more explicitly state some of the key caveats in the experimental design (e.g. using mesh bags, only measuring these variables once a year in September, measuring CO₂ production only once, and after harvesting).

I include line edits below, which raise some other key points

78: Can you explain in a few words why they might shift? You cite the Crowther et al 2016 paper, so perhaps mention that soil C losses under climate warming may lead boreal forests to shift from a carbon sink to carbon source.

82: Can you briefly differentiate why the boreal forest ecosystem, as compared to other forest ecosystems, is N limited? You state that a large fraction of soil N is bound to minerals and polyphenols. Why would this be different from other forests? This will be useful to place your question and results in contexts of other systems, which may or may not be N limited as well.

83: I would make this a little more clear to readers who may not be familiar with 'stable' SOM – i.e. explicitly state that again that this complexation of N compounds to minerals is what makes the N pool 'chemically stable'. Then you could say that one technique for measuring this stable N pool at this (with comes with its own flaws, as any protocol does) is by acid hydrolysis. But I would first make clear that you are referring to a protected, mineral-bound pool. Later, you can state that the way of measuring this 'stable' pool is through acid hydrolysis. An acid-unhydrolyzable pool doesn't mean anything substantive biologically 'in the field' – it is just a technique we have for trying to assess 'stable SOM.'

88: Do you mean transformations of SOM in boreal systems specifically? If not, what about AMF? If yes, then state you are referring specifically to boreal forests.

90: 'microorganisms and plant roots' seems a bit redundant, as you have just stated EEM and bacteria, both of which are microorganisms that are (or can be) associated with plant roots.

93: say "(referred to as the rhizosphere priming effect)."

95: say "(a phenomenon known as the Gadgil effect)"

99: add "are stabilized in soil on mineral surfaces and/or within soil aggregates"

107: "The study aims to elucidate the mechanisms underlying SOM transformations." I don't feel that the study designed precisely to look at SOM transformations. To do this I think would direct characterization of SOM, in specific C pools (i.e. particulate OM, mineral-associated SOM) as well as using some kind of characterization technique (i.e. NMR), or at least a stabilization assay that directly looked at stable/unstable SOM fractions. Rather, this paper is looking at the presence/absence of roots and mycorrhizae on the total SOM pool (measured through LOI), and on the 'chemically stable N' pool, measured by subtracting acid hydrolysable N from total N. Again, I think these results are useful, just the language needs to be a bit more accurate.

110: You hypothesize that plant roots and hyphae will stimulate SOM decomposition by stimulating microbial activity and the depletion of the stable N pool. Can you provide more background on these hypotheses and some justifications? What about the literature that shows that, despite higher rates of SOM mineralization, there is consistently higher total SOM in rhizosphere relative to bulk soil (i.e. Finzi et al GCB 2015)? You mention the co-occurring process of stabilization earlier in the manuscript, but this does not seem to be very fleshed out in your hypotheses. This seems like an important thing to include. Also, how might stabilization differ in the presence of roots + hyphae, versus hyphae alone?

113: "unsterilized, homogenized humus" – can you briefly explain if this was from the same site as the sites you placed the mesh bags? Were the mesh bags placed in the mineral soil, or on the soil surface, or in the organic horizon? A few more details here would be good, even though you describe this more in the Methods.

118: What about SOM formation, in addition to loss? You highlighted this as one of the main aims of the experiment, and one of the big gaps in current knowledge.

124: I'm not sure 'time of exposure' is the right phrase, unless you mean exposure to the mesh bag? You are considering the 1000 um mesh bag to be 'natural conditions,' so not sure exposure is right here. Maybe just say 'time'?

124: What are these statistics related to specifically? This sentence describes several chemical and biological properties as being significantly affected.

125: If chemical and biological properties are affected, how can you separate out what is due to absence of roots and/or hyphae, versus other conditions that are being changed with the addition of mesh bags (i.e. absence of soil fauna?). This seems like an important caveat that your study cannot parse out, and is worth mentioning this, and any other issues associated with size-exclusion mesh bags.

127: How are you measuring the relative speed of loss between years?

129: But what about other studies that find net increase of SOM in the rhizosphere, despite elevated decomposition rate? I.e. see Finzi et al. 2015 GCB for a metaanalysis. This needs to be addressed.

130: the rhizosphere priming effect. Also, are there some ecosystem contexts where the rhizosphere shows increases of SOM, whereas other systems where the rhizosphere shows depleted SOM? It would be good to place this in context of studies where rhizosphere has higher SOM than bulk soil (see comment above as well).

132: Can you reference a figure, and out a supplementary data table here (and statistics) when you say that decomposition was not enhanced in the 50 um treatment?

133-134: The wording is a little strange here. I don't get the main point you are trying to make.

139: Here you say 'duration of experiment' instead of 'time since exposure'. I'd be consistent with the terminology.

140: Was there a significant time x treatment interaction? If not, I don't think it makes sense to do post hoc tests and say that certain microbial groups were more or less dominant through time. Perhaps I am missing something though ...

142: What do you think explains these observations?

145: root-associated mycorrhizal fungi is redundant, don't need to include 'root-associated'

145: Dominate the organic soil layer relative to what? In what ecosystems? Also, I would use 'horizon' and not 'layer.'

150: Did you measure respiration throughout, or only at a single time point in the growing season? Is it likely you just did not capture the time at which CO₂ was being released? What other ways can you explain the reduction in SOM? Where else would it go?

156: I would add in the caveat here that 'for the methods we employed and at the timepoints we measured, microbial community structure did not explain changes in SOM decomposition.' There is evidence that changes during the actual growing season, which you did not capture (as well as changes in the active microbial groups – i.e. measured through stable isotope probing) would reveal changes.

Regardless, what does explain changes in SOM decomposition? Total microbial biomass? It has to be something. Do you report total microbial biomass anywhere? If not, this seems useful to report and to include in your models/figures from the PLFA data.

160: Can you state the value, or the change in value, and indicate beyond a P value which statistical test you are using here?

162: I'm not sure you can say that you measured increased stable SOM. You are only measuring acid hydrolysable N relative to total N, according to your methods. Why did you not just measure the acid unhydrolyzable N pool, and then say that 'chemically stable N increased'? It's referred to throughout as chemically stable N, and I think calling it the stable SOM pool is not the most accurate descriptor – there are other ways of more directly measuring stable SOM (including acid hydrolysis) – but it needs to go directly after the SOM pool, and actually measure the SOM that remains after some stabilization assay.

167: Maybe I'm missing something, but this would be an important correlation to show in a graph, perhaps in the main text.

170: If they did not significantly increase, then I would not include this result.

170 – 172: This is one potential interpretation, but it seems that it is based on this somewhat limited data (including data that was not significant), and it is a bit much to largely base a mechanistic framework around this.

175-176. This sentence does not flow clearly to me. Can you provide some background on what is known about ectomycorrhizal trees in boreal systems, and how these results show that ECM trees decrease N availability? I.e. more carefully spell this out.

194 – 201. This paragraph does not flow clearly. What I think would be useful here is a paragraph of various caveats of your experimental design, including where SOM may have gone, in addition to decomposition and uptake. If this paragraph is framed in this way, it would be much more clear why it is inserted here.

198: centennial means once every 100 years. I think you mean 'centurial'

204: This conclusion seems way too far reaching. Models have not been mentioned yet, and it would be useful to have a much more specific takeaway from this study that could be applied to improving 'plant soil microbial interactions' in biogeochemical models.

207: Can you provide values? How much did they increase decomposition and increase the stable N pool?

210: What does it specifically shed new light on. I think this conclusion paragraph needs to be re-written to be much more precise and specific as to what the experiment actually showed, and mention the caveats inherent in the design and approach.

220: Can you provide more details of the soil? Texture, pH, %C, %N, etc.?

224: Can you provide references that state that these exclusions effectively prohibit the entry of roots and hyphae? Also, I know many experiments have issues of root and hyphal penetration into these mesh sizes, even though they are meant to exclude them. Did you do anything to confirm that roots and hyphae did not penetrate? I.e. did you verify?

225: What depth? Why did you do between organic and mineral? Was it in the organic or mineral, or half and half?

227: If you are studying CO₂ release from the mesh bags after harvesting, you are not capturing what is happening in situ. I.e. you stated in your Results/Discussion that CO₂ mineralization was not elevated in the 1000 um treatment, but you may have simply not captured the period of time that CO₂ was being released. This is a major caveat you should discuss and mention.

237: It would be very useful to say that SOM was measured through loss on ignition here. It says in the supplementary, but I was confused, and it would be better to not have to dig around.

229: I'm not familiar with CO₂ respiration assays where soils are incubated at such a cold temperature. Can you explain why you did this, or provide a reference?

267: Did you do a Tukey's post hoc on the full model (i.e. all time points)? If so, how are you comparing the relative speed of SOM decay, as you describe in your Results?

It seems from your results you did two planned contrasts? Comparing treatments within the same year and also comparing the same treatment across years? Maybe make this more explicit if this is the case? What was the criteria by which you chose to do post-hoc Tukey's contrasts? If there was a significant Time*Treatment interaction? Or just a significant effect of Time or Treatment?

268: How was data transformed?

273: What's the 'experimental area'? Not sure what this means.

284: What do you mean by 'systematic changes'? What do you mean 'were masked by the differences in time'? Not sure what you mean here. Is there a reference you can cite for normalizing PLFA from 0-1. A few more details here would be good.

289: Reference?

466: Considering you only have 3 figures, this seems to be a very packed figure with too many subpanels. Can you separate into two separate figures?

476: Can you explain this – did you do two separate Tukey comparisons then? This was not clear to me in your Methods.

Response to Referees Letter

Plant roots increase decomposition but also the formation of stable soil organic matter in boreal forest soil

Bartosz Adamczyk, Outi-Maaria Sietiö, Petra Straková, Judith Prommer, Birgit Wild, Marleena Hagner, Mari Pihlatie, Hannu Fritze, Andreas Richter, Jussi Heinonsalo

(Comments marked in italics)

Reviewer #1 (Remarks to the Author):

The manuscript “Plant roots increase decomposition but also formation of stable soil organic matter in boreal forest soils” aims to contribute to mechanistically understand SOM transformation and N stabilization in forest humus by studying soil biological and chemical parameters in the absence or presence of roots and ectomycorrhizal fungal hyphae. This is an important and generally interesting scientific aim addressing several concepts related to the role of microbiota in soil organic matter transformation: rhizosphere priming vs. Gadgil effect as well as the emerging of the microbial carbon pump explaining SOM decomposition and stabilization. The broader context is a better understanding of C-transformation in boreal coniferous forest soils – a major C-sink in the northern hemisphere and thus an important factor in global C-budgeting.

The results base on a three year study using a soil mesh-bag experiment with different mesh sizes to prevent ingrowth of hyphae or roots or both. The mesh-bag samples of every year were analyzed using a suite of techniques that allow quantitation of different processes connected to SOM mineralization, turnover and stabilization as well as the link to N-mobilization and stabilization. The major results are that there were clear effects of mesh size on the dynamics and outcome of C-transformations and that the mesh size allowing in-growth of roots resulted in a higher stabilized SOM and organic N than when roots were excluded. The excellent and complementary chemical analyses on C and N-dynamics showed this convincingly.

However, there are three major points in the study that I find missing to accept the conclusion that was drawn from this study.

*1. My major concern is that the possible role of soil fauna is not considered at all: neither studied nor discussed - although one could easily argue that this study is about the impact of excluding or including soil mesofauna as well as on root exclusion vs. inclusion. Do the authors find the faunal component irrelevant for their concept, and if so why? Literature on the importance of soil (meso)fauna on SOM turnover and N dynamics is easy to access (for example: Setälä, H., Marshall, V. G., and Trofymow, J. A 1996. Influence of body size of soil fauna on litter decomposition and 15N uptake by poplar in a pot trial. *Soil Biol. Biochem.* 28, 1661–1675). Further hints can be found in a recent review Briones MJI 2018. *The Serendipitous Value of Soil Fauna in Ecosystem Functioning: The Unexplained Explained. Frontiers in Environmental Science*, 6(149)). Ideally, these missing data / information are included in a revised version.*

Answer: We agree with the Reviewer on the importance of soil fauna for decomposition and that this has not been sufficiently addressed in the previous version of the manuscript. We now added soil fauna data to Supplementary Table 2. Soil fauna was present in all mesh size treatments since we used non-sterile soil and the volume of the bags was large enough for the fauna to survive throughout the experiment. There was no major difference between treatments. The soil fauna data and their implications for this study are now discussed in lines 156 and 242, and mentioned also in Introduction line 96.

2. There is no evidence given that ectomycorrhizal hyphae were present in the 10 µm mesh bags. Although I find PLFA data useful for quantitative numbers and in this aspect superior to DNA-based

methods, in this specific case, DNA-based (amplicon sequencing) analyses are necessary to proof presence / absence of ectomycorrhizal taxa in the different mesh-size bags. This information could also be used to estimate presence and absence of different functional guilds of fungi i.e. saprotrophs and mycorrhizal fungi.

Answer: The microbial community composition in the same bags was investigated in detail and such analysis is included in the PhD thesis of one of the co-authors (ref: PhD thesis of Outi-Maaria Sietiö; <http://urn.fi/URN:ISBN:978-951-51-4571-0>). Since this dataset is very large, we did not include it in this manuscript in detail. However, we followed the Reviewer's suggestion and added the ECM and SAP ratios derived from the DNA based analysis (Supplementary Fig 2). As we used non-sterilized soil, DNAs of both guilds, mycorrhizal and saprotrophic fungi were present in all mesh bags. These data are discussed in line 153.

3. In the same line of thinking I was missing data on root biomass in the 1 mm mesh-bags which would add evidence on the importance of roots.

Answer: We also added data on root biomass (Supplementary Table 1). The results show no ingrowth into 1 and 50µm bags; in 1000µm root biomass was growing in time. Line 155.

Apart from the general concerns above, the strong positive aspects of the study are the chemical analytical methods and the results thereof which are of high quality and strongly contributing to new insights in SOM decomposition / stabilization in coniferous forest soils. The presentation of the results in the text and figures is generally very clear and convincing. Please find some detailed remarks on missing information and suggestions for improvements or clarification in the subsequent paragraph:

Abstract:

The abstract is clear and gives the major idea and results. However, the term "mechanistic framework" appears overselling in the light of not providing data on soil fauna or an estimate on ectomycorrhizal vs. saprotrophic fungal components.

Answer: We added information on ectomycorrhizal vs saprotrophic fungi and soil fauna to the manuscript. We further changed the term "mechanistic framework" into "a new framework" to avoid overstating.

Introduction:

line 111: "plant roots" seems too general. Consider writing "plant roots and/or mycorrhizae".

Answer: sentence modified accordingly; line 112.

Results and Discussion:

Line 124 onwards and Methods Summary line 270-272: the adonis function from vegan is described in the documentation as "robust alternative to parametric MANOVA". If the adonis function was used here, it would be better to write either "adonis" or "PERMANOVA" as vegan describes it. Instead of F-values, adjusted R² values would be more informative (again if adonis was used here).

Answer: We now present R²-values instead of F-values and corrected the term to "PERMANOVA", since the analysis is conducted with adonis using 999 permutations. Line 133, and also in summary of methods from line 326.

Line 127, 169: also for results that were not statistically significant but "tended to..." P-values should be given as an indication how strongly these tendencies are to be rated.

Answer: P-values were added as suggested. Line 136, 190.

Line 131: Why is this surprising? Is it realistic to assume that many mycorrhizal hyphae would grow into a root-free space? Here DNA analyses on fungi could give an indication on differences in the presence of ectomycorrhizal vs. decomposer fungi (see general comment 2 above).

Answer: This sentence was removed due to redundancy following suggestions of another reviewer. Note however that we added DNA data to the manuscript as suggested (Supplementary Figure 2).

Line 134: Citation 30: From the short sentence it is not clear which way of C-input via roots is addressed (rhizosphere deposits or dead root mass) and how the findings of Solly et al. on fine root turnover fit in the context of accelerating SOM decomposition in the present experiment.

Answer: yes, that is true, this sentence is not clear. We have modified it with no linkage to paper by Solly et al. (cited in another place, line 201).

Line 147: Has root biomass been determined in the 1000 µm mesh-bags? Was root ingrowth comparable among mesh-bags? Please add this information. In addition, biomass and identity of faunal components should be given in this paragraph.

Answer: See above - root and fauna data were added (Supplementary Table 1 and 2) and are shortly discussed in line 155.

Line 157: the claims regarding microbial community composition are based on PLFA which is fine for quantification of biomass but the resolution at the taxonomic is very low thus preventing any deeper insights into microbial community composition. Maybe this could be worded a bit more modestly: "community structure at the investigated level of PLFA... ()

Answer: We modified this part of text to underline that PLFA was used for microbial community structure (line 145). We have added also fungal DNA based results (Supplementary Table 2).

Line 167: please add which type of correlation was used (Pearson?).

Answer: yes, information was added, line 187.

Line 237: "soilC" space is missing: "soil C"

Answer: corrected, line 287.

Conclusion:

Apart from my concerns regarding the scientific part of the conclusions, the broader implications (line 210-213) are very generally worded and can be deleted without losing any information. I suggest to rethink the essence of the conclusion.

Answer: conclusions were re-written. In a revised version conclusions are improved by incorporation of broader implications, extrapolation to other ecosystems and by addition of directions of future studies. Lines 246-261.

Methods Summary

For adonis see comment above for line 124 onwards

Answer: We corrected the term to PERMANOVA, since the analysis was conducted with adonis using 999 permutations, and added the R2 values as suggested. Line 332.

Supplementary

Supplementary Figure 2: Reference 27 (line 214) should probably be 57 as in lines 292 and 502?

Answer: ref 27 in Suppl Information (now ref 29) is the same as 57 (now 59) in the main part of the text.

Supplementary Figure 4C: There seems to be a typo in the post-hoc designation a, ab, ab - one "ab" should be "b".

Answer: True, thank you! We have fixed it.

Reviewer #2 (Remarks to the Author):

Adamczyk et al. : Plant roots increase decomposition but also the formation of stable soil organic matter in boreal forest soil. This study potentially makes a very interesting and valuable contribution to the literature on mechanisms controlling SOM decomposition and stabilization in the soils of boreal forests. Warming climate is affecting C sequestration process and causing shifts in SOM accumulation and decomposition. Thus we are in urgent need for more empirical studies that provide fundamental step forward in our understanding of mechanisms of SOM stabilization and accumulation processes. I like very much the functional approach of this study, although I suspect there might be some treatment issues (mesh size) that may distort the results. The design of the experiment and the respective results raised the question: even though the root exclusion treatment (50µm) should allow only the hyphae to grow into the mesh bag, whether there were more of hyphal biomass compared to root and hyphae exclusion treatment (1 µm)? Fig 2c seems to confirm that there was no difference between the treatments (1000 µm, 50µm and 1 µm) during the first year – fungal PLFA was high in all treatments, also roots and fungal hyphae excluding (1 µm) mesh bags (still saprotrophs or old mycorrhizal signal, because the soil was not sterilized?); during the second and third year the in-growth of fungi into mesh bags with net size 50µm and 1 µm were similar (50µm and 1 µm seem to highly correlate, in Fig 1), while only the latter was meant to prevent hyphal in-growth. My question is can the authors confirm higher hyphal in-growth in root exclusion treatment (50µm) or the growth of (mainly) mycorrhizal hyphae decreased without roots during second year similarly to 1 µm mesh bags?

Answer: First: Yes, the soil was not sterilized to better reflect the natural conditions. This is why fungi were present also in the 1µm treatment, although such net size blocks fungal ingrowth. Moreover, we have to note that fungal PLFA markers are not a good proxy for fungal biomass since biomass is not well correlated with PLFA markers, and since the PLFA specific for fungi has also other origins, i.e. it is found in plants (e.g. Zelles 1997, Chemosphere 35:275-294). We have added ergosterol concentrations; ergosterol is a better indicator of fungal biomass since it is present only in fungi. The ergosterol data clearly showed higher amounts of fungi in the 50µm compared to the 1µm treatment (Supplementary Table 2). We further emphasize that microbial biomass and community composition inside the different mesh bags does not reveal their activity and functions. The chemistry and enzyme activity -based approach was used here to study the functionality of the microbial community inside the mesh bags. Changes in lines 144-159.

Line 49: does “exclusion of roots” mean presence of hyphae? This question relates to my questions about design (see above).

Answer: Yes, exclusion of roots means here presence of hyphae. We clarified it in text, line 49.

Line 51: concentration of organic...what?

Answer: N, we added it, line 52.

Lines 131-134: the statements here relate to my comments above. The authors state that C flow via mycorrhizal hyphae was not restricted in 50µm, but fungal PLFA values show lower in-growth since second year. I would ask the authors discuss this issue.

Answer: See above. We now added ergosterol data to better reflect fungal biomass than PLFA markers that can be found also in other organisms. Lines 144-159.

140-141: fungi were quite dominant already in the first year.

Answer: We agree; however, we now clarified the text that fungi only between treatments but within the same year are compared here. line 148

Line 184: Fig 2i should be Fig 2j

Answer: Fixed, but as we reorganized the figures according to comment of Reviewer 3, these figure panels can now be found in Fig. 3.

Lines 194-201: This is important and topic-related section of text, but feels a bit stand-alone here. The experiment was carried out in upper organic layer of the soil and DOC leaching effects are like out of frame, although I agree that DOC leaching from boreal forest soil is marginal and DOC has an important role in stabilization process of SOM in deeper layers. Can the authors make stronger link between their results and DOC leaching?

Answer: We agree and extended the discussion of possible caveats of the experimental design and the fate of C from mesh bags. Lines 233-243.

Line 212: Phrasing is not good: 'increasing the systems understanding' is not clear, what systems?

Answer: "systems" changed into "understanding of boreal forest", line 252.

However, this is very well written manuscript with clear message. Most novel and innovative are the results explaining the mechanisms underlying the stabilization of SOM (line 171-172) via differences in N-economy. And I find the framework presented on Fig 3 to be intriguing for further discussions on soil C cycle. Well done!

Answer: Thanks!

Reviewer #3 (Remarks to the Author):

In this manuscript, Adamczyk et al. aim to "elucidate the mechanisms underlying SOM transformations and its controls, focusing on the rhizosphere," specifically in a boreal forest ecosystem context. The authors used a three year field experiment, which involved different sized mesh bags that either excluded roots and hyphae (1 um mesh), excluded roots only (50 um), or did not exclude roots or hyphae (1000 um). They found that decomposition of SOM was elevated in the presence of roots + hyphae (1000 um) relative to the two other treatments, while the 'chemically stable N pool' increased in this treatment by 3, relative to the other treatments. They aim to use their results to present a mechanistic framework for how roots affect SOM decomposition and stabilization.

This study is a useful effort to try to untangle the role of roots and hyphae in N cycling (though for C cycling, which the authors also frequently mention, it would have been useful to have more detailed data on soil carbon). The role of roots and hyphae in the cycling of N and SOM is an understudied subject, especially in boreal systems, and it is particularly useful to have data over multiple years using a manipulative field design. The authors measured a wide suite of response variables, to provide a complete story, and these results definitely warrant publication. However, I have some concerns as to how the results are framed in terms of advancing understanding around the emerging idea of microbes for their role in stabilizing and decomposing SOM, as I'm not sure the study exactly addresses this point. Rather, it looks specifically at hydrolysable versus non hydrolysable N, but not the SOM pool. I find the data most compelling to provide a story of N dynamics, and not as much around SOM or C. Overall, I find the conceptual framework to be a bit of an overreach, based on the results of the experiment. While the results point to some interesting questions to follow up on, I do not feel they provide conclusive mechanistic evidence to advance such a mechanistic framework. For example, it seems to me that a large part of the justification for the framework is based on an observed correlation between condensed tannins and the size of the stable N pool. I would prefer the paper include more figures/tables that present actual data (at this point, only two data figures are included). That said, I commend the authors on conducting this important study, and advise that the take-home messages from the study be articulated a little more carefully and clearly – I found the last two paragraphs of the Discussion a bit hard to follow. One broad point is that I would like to see in the Discussion these results discussed in the context of boreal systems, and how different dynamics may play out in other forest ecosystems. Also, I would like to see the authors more persuasively state why these findings in this study are novel, to warrant publication in Nature Communications. The third paragraph of the Introduction states the aim of the paper, but I felt it did not persuasively state

why the hypotheses are unique, in the context of prior literature on this subject, and does not give background as to why the questions and approach are very novel. Last, I'd also like to see the authors more explicitly state some of the key caveats in the experimental design (e.g. using mesh bags, only measuring these variables once a year in September, measuring CO2 production only once, and after harvesting).

Answer: At first – thank you for valuable comments! We took all these comments into account, more precisely: - we re-framed the text, underlining that we concentrate more on SOM-N than SOM; we changed “mechanistic framework” into “new framework”, - we added one more fig (2d) to explain better correlation between stable SOM-N and CT; - we divided fig 2 into two figures (Fig 2 and 3), - we re-written two last paragraphs of discussion (including last one which now includes caveat of mesh-bag studies; lines 233-243) and we re-written hypotheses to underline novelty and possible formation of stable SOM (lines 111-118).

I include line edits below, which raise some other key points

78: Can you explain in a few words why they might shift? You cite the Crowther et al 2016 paper, so perhaps mention that soil C losses under climate warming may lead boreal forests to shift from a carbon sink to carbon source.

Answer: Done; We modified the sentence into “...with climate change SOM decomposition might increase shifting boreal forests from C sinks to C sources, thereby accelerating global warming¹..” line 78-79.

82: Can you briefly differentiate why the boreal forest ecosystem, as compared to other forest ecosystems, is N limited? You state that a large fraction of soil N is bound to minerals and polyphenols. Why would this be different from other forests? This will be useful to place your question and results in contexts of other systems, which may or may not be N limited as well.

Answer: Thank you for this valuable comment. We agree that “N-limitation” might not be the correct term so we have changed it into “low N availability” (New Phytologist 2016, 210:1165-1168; Terrer et al. 2017 New Phytologist) (line 45). Following Gill and Finzi (2016, Ecol Let), boreal forests have 13 fold greater C cost of N acquisition compared to that of tropical forests – we have included this fact into conclusions (line 258). Moreover, boreal forests often differ in this way that there is very low N deposition, which affects nutrient availability of other forest ecosystems more.

83: I would make this a little more clear to readers who may not be familiar with ‘stable’ SOM – i.e. explicitly state that again that this complexation of N compounds to minerals is what makes the N pool ‘chemically stable’. Then you could say that one technique for measuring this stable N pool at this (with comes with its own flaws, as any protocol does) is by acid hydrolysis. But I would first make clear that you are referring to a protected, N-bound pool. Later, you can state that the way of measuring this ‘stable’ pool is through acid hydrolysis. An acid-unhydrolyzable pool doesn’t mean anything substantive biologically ‘in the field’– it is just a technique we have for trying to assess ‘stable SOM.’

Answer: Stabilization of SOM by sorption to minerals is typical for soils with relatively low level of organic matter, i.e. with high level of minerals. However, mechanisms of stabilization are likely to be very different in purely organic mor layer. Recent studies in boreal forests highlight enzymatic oxidation and fungal communities as regulators of organic matter accumulation in the mor layer (Clemmensen et al. 2015, Kyaschenko et al. 2017, Stendahl et al 2017; Adamczyk et al. 2019). As concluded by Meier, Finzi and Phillips (Soil Biol Biochem, 2017): “In forest soils, this slow-cycling pool of SOM contains abundant quantities of organic N in the form of phenol-protein complexes” which underlines different mechanistic route of stabilization than minerals in boreal forest soils (at least in organic layer).

88: *Do you mean transformations of SOM in boreal systems specifically? If not, what about AMF? If yes, then state you are referring specifically to boreal forests.*

Answer: We clarified that we refer specifically to boreal forests (with almost no arbuscular mycorrhiza). Line 88.

90: *'microorganisms and plant roots' seems a bit redundant, as you have just stated EEM and bacteria, both of which are microorganisms that are (or can be) associated with plant roots.*

Answer: We removed "microorganisms and plant roots" from this sentence. Line 90.

93: *say "(referred to as the rhizosphere priming effect)."*

95: *say "(a phenomenon known as the Gadgil effect)"*

Answer: both done, lines 92 and 95.

99: *add "are stabilized in soil on mineral surfaces and/or within soil aggregates"*

Answer: added, lines 101-102.

107: *"The study aims to elucidate the mechanisms underlying SOM transformations." I don't feel that the study designed precisely to look at SOM transformations. To do this I think would direct characterization of SOM, in specific C pools (i.e. particulate OM, mineral-associated SOM) as well as using some kind of characterization technique (i.e. NMR), or at least a stabilization assay that directly looked at stable/unstable SOM fractions. Rather, this paper is looking at the presence/absence of roots and mycorrhizae on the total SOM pool (measured through LOI), and on the 'chemically stable N' pool, measured by subtracting acid hydrolysable N from total N. Again, I think these results are useful, just the language needs to be a bit more accurate.*

Answer: As study was done in highly organic layer of boreal forest soil, methods concentrating on minerals would not be appropriate. However, we understand that the term "SOM" is maybe not a correct one, so we re-directed the text more into SOM-N, as our work concentrate especially on N. Moreover, soil was characterized with FTIR, and main detected differences were in polyphenols and polypeptides. We added a sentence about it, line 184.

110: *You hypothesize that plant roots and hyphae will stimulate SOM decomposition by stimulating microbial activity and the depletion of the stable N pool. Can you provide more background on these hypotheses and some justifications? What about the literature that shows that, despite higher rates of SOM mineralization, there is consistently higher total SOM in rhizosphere relative to bulk soil (i.e. Finzi et al GCB 2015)? You mention the co-occurring process of stabilization earlier in the manuscript, but this does not seem to be very fleshed out in your hypotheses. This seems like an important thing to include. Also, how might stabilization differ in the presence of roots + hyphae, versus hyphae alone?*

Answer: We agree with the Reviewer and modified the hypotheses to take into account also the formation of stable SOM-N. We further extended the introduction to better outline the justification for our hypotheses (lines 106-7, for hypotheses 111-118)

113: *"unsterilized, homogenized humus" – can you briefly explain if this was from the same site as the sites you placed the mesh bags? Were the mesh bags placed in the mineral soil, or on the soil surface, or in the organic horizon? A few more details here would be good, even though you describe this more in the Methods.*

Answer: yes, we took soil from the same site and after placing it into bags we returned it into organic horizon. Needed modifications added in lines 120-122.

118: *What about SOM formation, in addition to loss? You highlighted this as one of the main aims of the experiment, and one of the big gaps in current knowledge.*

Answer: See above; we modified the hypotheses to consider also SOM formation, lines 111-118.

124: I'm not sure 'time of exposure' is the right phrase, unless you mean exposure to the mesh bag? You are considering the 1000 um mesh bag to be 'natural conditions,' so not sure exposure is right here. Maybe just say 'time'?

Answer: Changed into "time", line 132.

124: What are these statistics related to specifically? This sentence describes several chemical and biological properties as being significantly affected.

Answer: We linked it more clearly with figure 1, as these properties are explained in fig 1. Specifically, especially soil CT content, stable SOM-N, PLFA, and mass loss were affected by treatments.

125: If chemical and biological properties are affected, how can you separate out what is due to absence of roots and/or hyphae, versus other conditions that are being changed with the addition of mesh bags (i.e. absence of soil fauna?). This seems like an important caveat that your study cannot parse out, and is worth mentioning this, and any other issues associated with size-exclusion mesh bags.

Answer: We added a discussion of limitations of our approach to the manuscript (lines 233-243), as suggested in later comments. Note also that data on soil fauna have been added to the manuscript (Supplementary Table 2, lines 157, 243).

127: How are you measuring the relative speed of loss between years?

Answer: this question is unclear, as we are not writing about "speed of loss" in line 127. If the question is about "fastest" in line 126, we replaced it with "highest", line 134.

129: But what about other studies that find net increase of SOM in the rhizosphere, despite elevated decomposition rate? I.e. see Finzi et al. 2015 GCB for a metaanalysis. This needs to be addressed.

130: the rhizosphere priming effect. Also, are there some ecosystem contexts where the rhizosphere shows increases of SOM, whereas other systems where the rhizosphere shows depleted SOM? It would be good to place this in context of studies where rhizosphere has higher SOM than bulk soil (see comment above as well).

Answer: We added information about differences in SOM decomposition between rhizosphere and bulk soil citing meta-analysis by Finzi et al. 2015 GCB. Line 140-141; see also answer to line 110.

132: Can you reference a figure, and out a supplementary data table here (and statistics) when you say that decomposition was not enhanced in the 50 um treatment?

Answer: This sentence was removed as it was redundant.

133-134: The wording is a little strange here. I don't get the main point you are trying to make.

Answer: during modification of this part of text this sentence was removed.

139: Here you say 'duration of experiment' instead of 'time since exposure'. I'd be consistent with the terminology.

Answer: yes, we will stick to "time of exposure". Line 145.

140: Was there a significant time x treatment interaction? If not, I don't think it makes sense to do post hoc tests and say that certain microbial groups were more or less dominant through time. Perhaps I am missing something though ...

Answer: The sampling year had the strongest effect on soil microbial community structure, most likely driven by differences in environmental conditions between the growing seasons. To test for

treatment-effects over the years, we added the sampling year to the adonis as grouping factor (strata) since we treated the years as random variables.

142: What do you think explains these observations?

Answer: priming may be responsible here, we added this information, and we have added also citation to important meta-analysis by Finzi et al. GBC from 2015. Line 152.

145: root-associated mycorrhizal fungi is redundant, don't need to include 'root-associated'

Answer: true; however, this sentence was removed during modification of the text.

145: Dominate the organic soil layer relative to what? In what ecosystems?

Answer: ECM dominate over saprotrophic fungi in Oh and Of layers of boreal forest ecosystems; however, this sentence was removed during modification of the text.

150: Did you measure respiration throughout, or only at a single time point in the growing season? Is it likely you just did not capture the time at which CO₂ was being released? What other ways can you explain the reduction in SOM? Where else would it go?

Answer: Yes, CO₂ was measured only at a single point as it was technically not possible to measure CO₂ production continuously throughout the growing season. We consequently use the CO₂ data only to compare potential CO₂ production at harvesting point and not to “catch all released CO₂”. We now clarify this in lines 172-177. Overall, the reduction of SOM might have been due to decomposition and uptake by plant roots (note that forest plants have been shown to take up organic N which also contains C), and to some extent, also due to leaching (see last paragraph of the Discussion, lines 233-243).

156: I would add in the caveat here that 'for the methods we employed and at the timepoints we measured, microbial community structure did not explain changes in SOM decomposition.' There is evidence that changes during the actual growing season, which you did not capture (as well as changes in the active microbial groups – i.e. measured through stable isotope probing) would reveal changes. Regardless, what does explain changes in SOM decomposition? Total microbial biomass? It has to be something. Do you report total microbial biomass anywhere? If not, this seems useful to report and to include in your models/figures from the PLFA data.

Answer: We added similar sentence in lines 172-177. Moreover, we have added measurements of ergosterol content (fungal biomass) and DNA studies of fungi (sapro vs mycorrhizal fungi), lines 152-158.

160: Can you state the value, or the change in value, and indicate beyond a P value which statistical test you are using here?

Answer: This information was added, line 180-181. Here we have used ANOVA followed by Tukey post-hoc test.

162: I'm not sure you can say that you measured increased stable SOM. You are only measuring acid hydrolysable N relative to total N, according to your methods. Why did you not just measure the acid unhydrolyzable N pool, and then say that 'chemically stable N increased'? Its referred to throughout as chemically stable N, and I think calling it the stable SOM pool is not the most accurate descriptor – there are other ways of more directly measuring stable SOM (including acid hydrolysis) – but it needs to go directly after the SOM pool, and actually measure the SOM that remains after some stabilization assay.

Answer: We used method which is commonly used to measure stable SOM: for example, this method was used in Kallenbach et al. 2016, (<https://www.nature.com/articles/ncomms13630>). Secondly, question of reviewer suggest thinking about different types of soil, probably highly inorganic, very far from mor humus. As we have been working with highly organic soil, stabilization

assays mentioned by reviewer are not in use.

167: Maybe I'm missing something, but this would be an important correlation to show in a graph, perhaps in the main text.

Answer: Figure showing correlation between CT and stable SOM-N added to Fig 3.

170: If they did not significantly increase, then I would not include this result.

170 – 172: This is one potential interpretation, but it seems that it is based on this somewhat limited data (including data that was not significant), and it is a bit much to largely base a mechanistic framework around this.

Answer to both comments: We added a figure showing the significant correlation between stable SOM-N and CT (Fig 3) and clarified that the mechanism of fungal necromass stabilization was already demonstrated in our earlier paper (Adamczyk et al. 2019, New Phytologist, please see also Commentary to this paper by Hättenschwiller et al. 2019, New Phytologist). In this earlier paper we used fungal necromass instead of humus in mesh bags to prove such mechanism: In laboratory study with pine microcosms decomposition of the fungal necromass was decreased by the formation of complexes with tannins and in the field study demonstrated that fungal necromass–tannin complexes may be created also under natural conditions (Adamczyk et al. 2019); here, our correlations suggest that such a mechanism can have implications for SOM-N in general. Lines 190-207.

175-176. This sentence does not flow clearly to me. Can you provide some background on what is known about ectomycorrhizal trees in boreal systems, and how these results show that ECM trees decrease N availability? I.e. more carefully spell this out.

Answer: We apologize for being unclear and changed this and an earlier sentence, lines 203-207.

194 – 201. This paragraph does not flow clearly. What I think would be useful here is a paragraph of various caveats of your experimental design, including where SOM may have gone, in addition to decomposition and uptake. If this paragraph is framed in this way, it would be much more clear why it is inserted here.

Answer: Thank you for a great idea; we added a paragraph on caveats of the study design as suggested, lines 233-243.

198: centennial means once every 100 years. I think you mean 'centurial'

Answer: thank you! Corrected. Line 237.

204: This conclusion seems way too far reaching. Models have not been mentioned yet, and it would be useful to have a much more specific takeaway from this study that could be applied to improving 'plant soil microbial interactions' in biogeochemical models.

Answer: yes, it is true, modelling is not a good thing to mention here. We modified strongly this part. Lines 246-261.

207: Can you provide values? How much did they increase decomposition and increase the stable N pool?

Answer: These values are added a bit earlier, into main part of discussion, line 198.

210: What does it specifically shed new light on. I think this conclusion paragraph needs to be re-written to be much more precise and specific as to what the experiment actually showed, and mention the caveats inherent in the design and approach.

Answer: Caveats were discussed in the last paragraph of discussion and in lines 233-243. Conclusions were re-written. In a revised version conclusions are improved by incorporation of broader

implications, extrapolation to other ecosystems and by addition of directions of future studies. Lines 246-261.

220: Can you provide more details of the soil? Texture, pH, %C, %N, etc.?

Answer: added, line 266-267.

224: Can you provide references that state that these exclusions effectively prohibit the entry of roots and hyphae? Also, I know many experiments have issues of root and hyphal penetration into these mesh sizes, even though they are meant to exclude them. Did you do anything to confirm that roots and hyphae did not penetrate? I.e. did you verify?

Answer: These exclusions were based on papers like Wallander et al. 2001 (see line 474). Yes, it is true that sometimes bags are broken and roots can penetrate inside. We were always checking for root absence (1µm, 50µm)/presence (1mm; data of root presence added) as well as fungal ingrowth (50µm, 1mm; through added results of ergosterol). Data for root ingrowth and fungal biomass are gathered in Supplementary Table 1 and mentioned in the text (lines 152-158).

225: What depth? Why did you do between organic and mineral? Was it in the organic or mineral, or half and half?

Answer: Bags were placed between the organic and topmost mineral soil layer in the depth –of about 3-4cm (depending on the organic layer depth). Positioned in this way, the bags were in fact part of lower organic layer and exposed to roots and fungi. Bags were not within mineral layer. The thickness of organic layer was not large enough to allow burial of bags within organic layer. Line 273.

227: If you are studying CO₂ release from the mesh bags after harvesting, you are not capturing what is happening in situ. I.e. you stated in your Results/Discussion that CO₂ mineralization was not elevated in the 1000 um treatment, but you may have simply not captured the period of time that CO₂ was being released. This is a major caveat you should discuss and mention.

Answer: We have added this important caveat into discussion, lines 172-178, not into last paragraph of discussion, as we felt that this caveat has to be discussed at place where CO₂ results are mentioned.

237: It would be very useful to say that SOM was measured through loss on ignition here. It says in the supplementary, but I was confused, and it would be better to not have to dig around.

Answer: Information added, line 287.

229: I'm not familiar with CO₂ respiration assays where soils are incubated at such a cold temperature. Can you explain why you did this, or provide a reference?

Answer: The purpose of the soil respiration measurement at cold temperature was to mimic soil conditions in the field at the time of sampling. The humus-bags were collected during autumn in September when soil temperatures had reached around +4 °C. We wanted to capture the respiration rate in the field conditions, and also to be able to link the CO₂ respiration to the other microbial and compound specific chemistry measured from the samples. Lines 277-286.

*267: Did you do a Tukey's post hoc on the full model (i.e. all time points)? If so, how are you comparing the relative speed of SOM decay, as you describe in your Results? It seems from your results you did two planned contrasts? Comparing treatments within the same year and also comparing the same treatment across years? Maybe make this more explicit if this is the case? What was the criteria by which you chose to do post-hoc Tukey's contrasts? If there was a significant Time*Treatment interaction? Or just a significant effect of Time or Treatment?*

Answer: Yes, there was significant time(year) x treatment interaction and yes, we compared treatments within the same year and the same treatment across years (so separate Tukeys). This

information was added to the MS. The Tukey test was chosen as we wanted to compare means to each other (inside the same year, or the same treatments between years) in a conservative way (i.e., so only really clear differences are visible, on the other hand e.g. LSD test is far less conservative. Lines 325-336.

268: How was data transformed?

Answer: Data were transformed using log10, as well as arcsin for percentage data; this information was added to line 327.

273: What's the 'experimental area'? Not sure what this means.

Answer: We replaced "area" with "site". The mesh bags were buried at three different sites ("experimental areas") in the same forest more than 50 m apart from each other. Line 341.

284: What do you mean by 'systematic changes'? What do you mean 'were masked by the differences in time'? Not sure what you mean here. Is there a reference you can cite for normalizing PLFA from 0-1. A few more details here would be good.

Answer: Since the sampling year had the most significant effect on the PLFA patterns and values, we normalized the heterogeneity between years between 0 and 1 with the decostand function from vegan prior to performing the LDA, in order to visualize the differences caused by the mesh treatments. We now clarify this in lines 345-349.

289: Reference?

Answer: Added, line 351, reference no 71.

466: Considering you only have 3 figures, this seems to be a very packed figure with too many subpanels. Can you separate into two separate figures?

Answer: Done as suggested, fig 2 divided into fig 2 and fig 3.

476: Can you explain this – did you do two separate Tukey comparisons then? This was not clear to me in your Methods.

Answer: Yes, we performed separate Tukeys. This information was added to the method summary, line 326-333.

REVIEWERS' COMMENTS:

Reviewer #1 (Remarks to the Author):

1. The possible role of soil mesofauna has now been sufficiently addressed. Thanks for including this information.
2. Data on ectomycorrhizal and saprotrophic fungal guilds are now given. Both data sets ECM and SAP show a decrease in 1 μm mesh size bags. I find the 5-fold higher arbitrary units of ECM compared to saprotrophs astonishing but this may be a methodological issue of primer sequencing. However, this is not relevant for the main assertion of fungal decrease in 1 μm meshbags. Thus, these additional data is sufficient.
3. Root ingrowth data are now included showing that the mesh-bag method worked and that roots are the relevant factor.
4. The conclusion is now more elaborate but I still have a question:
Line 251-252. You write that "The framework provided by our study may facilitate an improved understanding of boreal forest responses to global changes ..." How can the results of this study contribute to better understand "responses of boreal forests to climate change"? Should it rather be forest soils? Do not the results give much more indication of what consequences deforestation or forest die-back will have on forest soil biogeochemistry (than on forest responses)?
5. All other (minor) issues from my side are satisfyingly addressed.

Reviewer #2 (Remarks to the Author):

The improvements, particularly the addition of data of root growth, presence of fauna (Tables S1 and S2) and ECM/SAP ratio based on DNA analysis make this ms greatly better, more convincing and I truly believe that this piece of work is a valuable addition to the body of research in the field of SOM transformations and C and N stabilization in boreal forest soil.

I enjoyed the reading and I have just few comments: 1) in line 243 you make quite strong statement about caveats effects on the results. In case of soil fauna and moisture the original data support the statement as also the potential effect of mesh bag manipulation is mentioned, but in case of DOC leaching, the statement is based on the literature (Pumpanen et al 2014, ref 47), where the conclusion that DOC runoff from boreal soils is marginal has been made for undisturbed soils. But in this case the processing of soil (homogenization) can have an effect and we do not know its effect on different treatments. I would suggest to rephrase accordingly that in case of DOC leaching you just hypothesize that your samples are similar to situation that is measured for undisturbed forest soil.

2) please be consistent and keep 2 decimal places for both R2 and r: Line 132, 133, 146: the number of decimal places for presenting R2 values is too high, 2-3 digits would be enough, moreover, in line 190, r values are presented in two different ways, one with 2 decimal places the other with 3.

In conclusion, I would be happy to see this ms to be published in Nature Communications and to open wider discussion on SOM decomposition mechanisms.

Response to Referees Letter

Plant roots increase both decomposition and stable organic matter formation in boreal forest soil.

Bartosz Adamczyk, Outi-Maaria Sietiö, Petra Straková, Judith Prommer, Birgit Wild, Marleena Hagner, Mari Pihlatie, Hannu Fritze, Andreas Richter, Jussi Heinonsalo

REVIEWERS' COMMENTS:

Reviewer #1 (Remarks to the Author):

1. The possible role of soil mesofauna has now been sufficiently addressed. Thanks for including this information.
2. Data on ectomycorrhizal and saprotrophic fungal guilds are now given. Both data sets ECM and SAP show a decrease in 1 μm mesh size bags. I find the 5-fold higher arbitrary units of ECM compared to saprotrophs astonishing but this may be a methodological issue of primer sequencing. However, this is not relevant for the main assertion of fungal decrease in 1 μm meshbags. Thus, these additional data is sufficient.
3. Root ingrowth data are now included showing that the mesh-bag method worked and that roots are the relevant factor.
4. The conclusion is now more elaborate but I still have a question:
Line 251-252. You write that “The framework provided by our study may facilitate an improved understanding of boreal forest responses to global changes ...” How can the results of this study contribute to better understand “responses of boreal forests to climate change”? Should it rather be forest soils? Do not the results give much more indication of what consequences deforestation or forest die-back will have on forest soil biogeochemistry (than on forest responses)?
5. All other (minor) issues from my side are satisfyingly addressed.

Reviewer #2 (Remarks to the Author):

The improvements, particularly the addition of data of root growth, presence of fauna (Tables S1 and S2) and ECM/SAP ratio based on DNA analysis make this ms greatly better, more convincing and I truly believe that this piece of work is a valuable addition to the body of research in the field of SOM transformations and C and N stabilization in boreal forest soil.

I enjoyed the reading and I have just few comments: 1) in line 243 you make quite strong statement about caveats effects on the results. In case of soil fauna and moisture the original data support the statement as also the potential effect of mesh bag manipulation is mentioned, but in case of DOC leaching, the statement is based on the literature (Pumpanen et al 2014, ref 47), where the conclusion that DOC runoff from boreal soils is marginal has been made for undisturbed soils. But in this case the processing of soil (homogenization) can have an effect and we do not know its effect on different treatments. I would suggest to rephrase accordingly that in case of DOC leaching you just hypothesize that your samples are similar to situation that is measured for undisturbed forest soil. 2) please be consistent and keep 2 decimal places for both R² and r: Line 132, 133, 146: the number of decimal places for presenting R² values is too high, 2-3 digits would be enough, moreover, in line 190, r values are presented in two different ways, one with 2 decimal places the other with 3. In conclusion, I would be happy to see this ms to be published in Nature Communications and to open wider discussion on SOM decomposition mechanisms.

Reply to reviewers

Reviewer #1 (Remarks to the Author):

- 1. The possible role of soil mesofauna has now been sufficiently addressed. Thanks for including this information.*
- 2. Data on ectomycorrhizal and saprotrophic fungal guilds are now given. Both data sets ECM and SAP show a decrease in 1 μm mesh size bags. I find the 5-fold higher arbitrary units of ECM compared to saprotrophs astonishing but this may be a methodological issue of primer sequencing. However, this is not relevant for the main assertion of fungal decrease in 1 μm meshbags. Thus, these additional data is sufficient.*
- 3. Root ingrowth data are now included showing that the mesh-bag method worked and that roots are the relevant factor.*

Thank you, we also believe that including information concerning ECM and SAP, soil fauna and roots improved the manuscript.

- 4. The conclusion is now more elaborate but I still have a question:
Line 251-252. You write that "The framework provided by our study may facilitate an improved understanding of boreal forest responses to global changes ...". How can the results of this study contribute to better understand "responses of boreal forests to climate change"? Should it rather be forest soils? Do not the results give much more indication of what consequences deforestation or forest die-back will have on forest soil biogeochemistry (than on forest responses)?*

We strongly agree here. We replaced "boreal forests" with "forests soils" according to reviewer suggestion. Yes, definitely our MS indicates possible consequences of deforestation and its effect of forest soil biochemistry.

- 5. All other (minor) issues from my side are satisfyingly addressed.*

Thank you.

Reviewer #2 (Remarks to the Author):

The improvements, particularly the addition of data of root growth, presence of fauna (Tables S1 and S2) and ECM/SAP ratio based on DNA analysis make this ms greatly better, more convincing and I truly believe that this piece of work is a valuable addition to the body of research in the field of SOM transformations and C and N stabilization in boreal forest soil. I enjoyed the reading and I have just few comments: 1) in line 243 you make quite strong statement about caveats effects on the results. In case of soil fauna and moisture the original data support the statement as also the potential effect of mesh bag manipulation is mentioned, but in case of DOC leaching, the statement is based on the literature (Pumpanen et al 2014, ref 47), where the conclusion that DOC runoff from boreal soils is marginal has been made for undisturbed soils. But in this case the processing of soil (homogenization) can have an effect and we do not know its effect on different treatments. I would suggest to rephrase accordingly that in case of DOC leaching you just hypothesize that your samples are similar to situation that is measured for undisturbed forest soil.

We have followed important suggestion of Reviewer and modified the sentence about DOC accordingly.

2) please be consistent and keep 2 decimal places for both R2 and r: Line 132, 133, 146: the number of decimal places for presenting R2 values is too high, 2-3 digits would be enough, moreover, in line 190, r values are presented in two different ways, one with 2 decimal places the other with 3. In conclusion, I would be happy to see this ms to be published in Nature Communications and to open wider discussion on SOM decomposition mechanisms.

All needed corrections provided in revised version of manuscript.